



# Multi-objective optimisation of a rock coast evolution model with cosmogenic [10]Be analysis for the quantification of long-term cliff retreat rates

Jennifer R. Shadrick[1], Martin D. Hurst[2], Matthew D. Piggott[1], Bethany G. Hebditch[1], Alexander J.
Seal[1], Klaus M. Wilcken[3], Dylan H. Rood[1]

[1]Earth Science and Engineering, Imperial College London, London, SW7 2AZ, United Kingdom
[2]School of Geographical and Earth Sciences, University of Glasgow, Glasgow, G12 8QQ, United Kingdom
[3]Institute for Environmental Research (IER), Australian Nuclear Science and Technology Organization (ANSTO), Lucas
Heights, NSW 2234, Australia

*Correspondence to*: Jennifer R. Shadrick (jrs17@ic.ac.uk)

**Abstract.** This paper presents a methodology that uses site-specific topographic and cosmogenic [10]Be data to perform multi-objective model optimisation of a coupled coastal evolution and cosmogenic radionuclide production model. Optimal parameter estimation of the coupled model minimises discrepancies between model simulations and measured data to reveal the most likely history of rock coast development. This new capability allows for a time-series of cliff retreat rates to be

quantified for rock coast sites over millennial timescales. This is the first study that has 1) applied a process-based coastal evolution model to quantify long-term cliff retreat rates for real, rock coast sites, and 2) coupled cosmogenic radionuclide analysis with a process-based model. The Dakota optimisation software toolkit is used as an interface between the coupled coastal evolution and cosmogenic radionuclide production model and optimisation libraries. This framework enables future applications of datasets associated with a range of rock coast settings to be explored. Process-based coastal evolution models

simplify erosional processes and, as a result, often have equifinality properties, for example, that similar topography develops via different evolutionary trajectories. Our results show that coupling modelled topography with modelled [10]Be concentrations can reduce equifinality in model outputs. Furthermore, our results reveal that multi-objective optimisation is essential in limiting model equifinality caused by parameter correlation to constrain best-fit model results for real-world sites. Results from two UK sites indicate that the rates of cliff retreat over millennial timescales are primarily driven by the

rates of relative sea level rise. These findings provide strong motivation for further studies that investigate the effect of past and future relative sea level rise on cliff retreat at other rock coast sites globally.



# 1 Introduction

Fundamental features of a rock coast are a sea cliff and shore platform, and the rate of cliff retreat is foremost the collective result of processes eroding the cliff face horizontally and the shore platform vertically (Sunamura, 1992; Trenhaile, 2008a). The ability to erode a cliff face depends fundamentally on the type of cliff material exposed to the delivery of energy to the cliff surface, usually in the form of waves. In turn, delivery of wave energy is mediated by the configuration of the shore platform, beach width, and wave climate (Sunamura, 1992). Thus, the processes that effect the weathering, erosion and

transport of shore platform, intact cliff, failed cliff and other beach material are an important part of the whole process of 'cliff erosion' (Coombes, 2014; Hurst et al., 2016; Limber and Murray, 2011; Masteller et al., 2020; Naylor and Stephenson, 2010; Thompson et al., 2019). These complex and varied processes make predicting long-term cliff erosion rates difficult. Erosional processes are governed by climate, relative sea level (RSL), tides and local lithology type and structure (Kennedy et al., 2014), which further complicate the prediction of large spatial and temporal-scale erosion rates at rock coast sites.

With climate change threatening the stability of these coastlines through RSL rise and increased storminess (Trenhaile, 2014), accurate long-term predictions of erosion rates will be highly valuable in the development of scenarios within the context of coastal management.

Understanding and quantifying the long-term trajectory of cliff erosion is central to the development of predictive coastal

evolution models that account for a changing climate. Current records of cliff retreat can only be observed through historical records, which are typically over a ~150 year time period (Brooks, 2010; Dornbusch et al., 2008). This time period is monopolised by engineering and modification of coastlines, hindering observations of their natural behaviours (Hurst et al., 2016). Furthermore, infrequent mass wasting events can obscure relationships between climate and average erosion rates in short-term records (Trenhaile, 2014). This means projections of cliff retreat derived solely from short-term data records can

be unreliable (Sunamura, 2015). It is critical that cliff retreat is studied over millennial timescales that are able to integrate changes in RSL rise, the return period of episodic erosion events and that precede the influence of anthropogenic modifications to the coastline.

The contribution of cosmogenic radionuclide (CRN) analysis to the advances in rock coast science are well known, but

further potential in its application to rock coasts is recognised (Trenhaile, 2018). The quantification of rock coast evolution is impeded by scarce and slow erosion indicators and a lack of dateable deposits (Trenhaile, 2008a). However, CRN analysis can be applied directly to the shore platform surface to calculate exposure time and erosion rates (Regard et al., 2012). Cosmic rays interact with target elements in the upper few metres of the Earth's surface to produce CRNs (Gosse and Phillips, 2001). Model predictions of CRN concentrations across a shore platform display a characteristic 'humped'

distribution profile across-shore (Hurst et al., 2017), for which the magnitude of the hump is inversely proportional to cliff retreat rate (Regard et al., 2012). Previous applications of CRN measurements on cliffs and shore platforms have been used





to quantify cliff retreat rates (Duguet et al., 2021; Hurst et al., 2016; Regard et al., 2012; Rogers et al., 2012; Swirad et al., 2020), understand Quaternary-scale shore platform exposure history (Choi et al., 2012), date major mass wasting events (Barlow et al., 2016; Recorbet et al., 2010) and constrain shore platform denudation rates (Raimbault et al., 2018).

Combining CRN analysis with a coastal evolution model can help reveal site-specific, long-term cliff retreat and shore platform lowering rates (Trenhaile, 2018).

A novel contribution here is the use of a morphodynamic model of rock coast development to interpret CRN concentrations. Furthermore, this study sees the first application of a morphodynamic rock coast evolution model to real-world sites in order

to model past cliff retreat rates. Coupling CRN concentrations with topography can help constrain modelled morphodynamics and replicate real-world sites. Equally, accurate morphodynamic development provided by the coastal evolution model is needed in order to interpret CRN concentrations. Application of a process-based model allows for replication of CRN production and regulation through time that corresponds directly to rock coast profile development. In order to apply a morphodynamic model to a real rock coast site and accurately model CRN concentrations, we need a

rigorous method of comparing model results with measured field data. Primarily, we need to establish whether the model is capable of replicating both the measured topography and CRN concentrations simultaneously to ensure the modelled cliff retreat rates are an accurate reflection of the evolutionary history at the rock coast sites in question.

In order to interpret CRN concentrations, a process-based model of rock coast development is required. Matsumoto et al.

(2016) presents an effective, exploratory coastal evolution model that simplifies wave properties. The model can produce a wide range of endmember across-shore profile shapes, and generally identify dominant erosion processes (Matsumoto et al., 2018). However, simplified processes and lack of field data calibration inhibit the application to real-world sites. As such, replication of an observed topographic profile of a cliff and shore platform has not yet been achieved. Equifinality is often an unavoidable property of modelled geomorphic systems and, as a result of simplified processes, causes the same endmember

results to be produced from non-distinctive parameter values. Previous explorations into the relative contributions of wave and weathering-driven erosion revealed evidence of equifinality in the model (Matsumoto et al., 2018). In particular, similar profile shapes were produced in mega-tidal settings when considering a significant range of wave force. The addition of modelled CRN concentrations to the topographic profiles has the potential to address equifinal model results. Moreover, field data calibration can be used to identify and constrain the conditions that cause equifinality, so that this abstract model

can be applied to real-world sites.

This study uses multiple site-specific datasets in order to calibrate a model that couples the Matsumoto et al. (2016) coastal evolution model and the Hurst et al. (2017) dynamic coastal evolution and cosmogenic radionuclide production model. We use Dakota optimisation software (Adams et al., 2019) with the Queso Bayes calibration library (Estacio-Hiroms et al.,

2016) to implement multi-objective model optimisation using the Metropolis-Hastings Markov Chain Monte Carlo (MCMC)



method (Hastings, 1970; Metropolis et al., 1953). We demonstrate that with our optimisation method, wave and weathering processes are adequately simulated so as to model real-world sites. The model was calibrated to a high-resolution, across-shore topographic profile and, more importantly, high precision $^{10}$Be concentrations. We are therefore able to extract further information from the model, such as: 1) the antiquity of the shore platform, 2) a time series of long-term cliff retreat rates

and 3) are able to distinguish between different forms of erosion acting on real-world profiles over a large temporal scale, while addressing and limiting model equifinality. This new capability allows us to better constrain the geomorphic history of rock coast evolution and input model parameters. As a result, the constrained model parameters can be used together with future RSL predictions to project rock coast profile development so that predictions of future coastal erosion rates are possible.

## 2 Field location overview


We utilise field datasets taken from two UK sites to develop an approach to calibrate the coastal evolution model (Fig. 1). The first study site at Bideford is located on Devon's north coast in south-west England. The study was carried out within the Bideford Formation, part of the Upper Carboniferous deposits of the Culm Basin in Devon, which is composed of nine coarsening-upwards cycles from black mudstone to massive sandstones (Edmonds et al., 1979). The beds are well exposed in

the cliff and platform and are steeply dipping (60-65°) to the SW and strike SE-NW roughly perpendicular to the coastline. The intertidal shore platform has a width of ~230 m and a near-continuous gradient (tan $\beta$) of 0.02. Cliff height reaches 36 m, with a ~24 m wide beach overlying the cliff-platform junction. The south-west coast of the UK is mega-tidal, with a mean spring tidal range of 8.41 m at the Bideford coastline (National Tidal and Sea Level Facility, 2021).

The second study site at Scalby, is located in north Yorkshire on the east coast of the UK. The Scalby site is located within the Mid-Jurassic Long Nab member of the Scalby Formation, which is comprised of fine-grained sandstone (Riding and Wright, 1989). The beds at Scalby are shallowly dipping (~12°) to the SE and strike SW-NE. At Scalby, the intertidal shore platform width reaches ~240 m, has a gently sloping gradient (tan $\beta$) of 0.01 and a steeply sloping 50-80 m-high coastal bluff is present. The east coast of the UK has a meso-macrotidal range half of that at the south-west site, with mean spring

tidal range of 4.6 m at the Scalby coastline (National Tidal and Sea Level Facility, 2021).



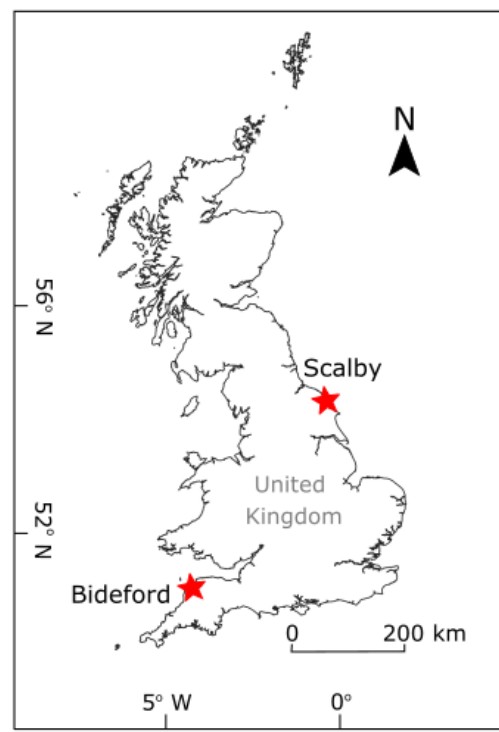


**Figure 1: Map of the United Kingdom with 2 rock coast sites labelled. Bideford located in south-west England on the north coast of Devon. Scalby located on the east coast of England in north Yorkshire.**

## 3 Methods

Using methods described below, we aim to quantify long-term, transient cliff retreat rates that will enable better predictions

of erosion rates at rock coast sites across the UK and world-wide. This flexible optimisation method implemented within the Dakota environment allows for simple replication with new datasets and can be applied to a range of rock coast settings.

### 3.1 Field datasets

Two distinct datasets are used to calibrate the coastal evolution model. The first dataset is an across-shore topographic profile (Fig. 2a). The profile is extracted from a high-resolution digital surface model (DSM) generated by structure-from-

motion analysis of aerial photographs collected by an unmanned aerial vehicle (UAV) survey at both sites. The second dataset is a $^{10}$Be concentration across-shore profile (Fig. 2b). In situ bedrock samples for CRN analysis were taken along an across-shore transect at ~10 m intervals from a sandstone bed at both sites. Analyses of $^{10}$Be/$^{9}$Be ratios using accelerator mass spectrometry (AMS) were carried out at the Australian Nuclear Science and Technology Organisation using the 6 MV Sirius tandem accelerator (Wilcken et al., 2017). Measured $^{10}$Be concentrations were normalised to the KN-5-3 standard with

an assumed ratio of 6.320 x $10^{-12}$ (t$_{1/2}$=1.36 Ma, (Nishiizumi et al., 2007)). Details of CRN sample collection and preparation, and drone survey data collection, processing and swath profile generation will be presented and interpreted in detail in future





work (Shadrick et al., in prep). In this study, our measured data serve solely as input test datasets for developing appropriate multi-objective optimisation routines, thus details of these test datasets are not central to this investigation.

Measured $^{10}$Be concentrations are corrected for both chemistry background and inherited levels of $^{10}$Be by subtracting the concentration of $^{10}$Be present in process blank samples and a shielded sample taken from a sea cave or cliff base respectively. Two key $^{10}$Be production pathways exist: $^{10}$Be produced from spallation reactions (4.0 atoms g$^{-1}$ yr$^{-1}$), normalised to sea level high latitude (SLHL), and muogenic-produced $^{10}$Be (0.028 atoms g$^{-1}$ yr$^{-1}$). $^{10}$Be production in the upper few metres of the Earth surface is dominated by exposure to secondary cosmic-ray neutrons (spallation), whereas

muon-produced $^{10}$Be prevails with greater depth below the Earth's surface owing to its longer attenuation length (42000 kg m$^{-2}$) in contrast to the spallation attenuation rate of 1600 kg m$^{-2}$ (Braucher et al., 2013). The production of *in situ* $^{10}$Be declines exponentially with depth below the Earth's surface as cosmic ray flux attenuates (Balco et al., 2008; Gosse and Phillips, 2001; Hurst et al., 2017; Mudd et al., 2016). Because the cliff/sea cave samples are previously shielded by ~40-80 m of rock (see section 2), the concentration of $^{10}$Be within the shielded sample is assumed to be entirely produced from deep-

penetrating muons, with no contributions accounted to neutron spallation through exposure to cosmic rays. Correcting shore-platform samples using the shielded samples corrects for any $^{10}$Be present in the rock before spallogenic $^{10}$Be becomes dominant. The exposure time is then calculated from the corrected $^{10}$Be concentrations. See supplementary materials Table S1 and Table S2 for $^{10}$Be concentrations used as model inputs for Bideford and Scalby sites.

A RSL history record from a glacial-isostatic adjustment (GIA) model (Bradley et al., 2011) shows a constant but declining rate of RSL rise across the Holocene for both sites (Fig. 2c). At both sites, the RSL 8,000 years BP was at an elevation of ~16 m lower than the present day RSL. At Scalby, average rates of RSL rise reduce from +0.7 mm y$^{-1}$ to +0.05 mm y$^{-1}$ across the last 8,000 yrs. Similarly, at Bideford, average rates of RSL rise reduce from +0.7 mm y$^{-1}$ to +0.04 mm yr$^{-1}$ across the last 10,000 years.






**Figure 2: Measured topographic profile (a) and** $^{10}$**Be concentration data (b) from Bideford and Scalby shore platform rock coast sites. Average elevation in the swath profile is shown by the solid line and uncertainty for the topographic profile shown by the shaded area sum the standard error from a linear regression of the topographic swath profile and the resolution of the UAV imagery (a).** $^{10}$**Be concentration values (b) are corrected for chemistry background using process blank samples and inherited** $^{10}$**Be using shielded cliff samples. Errors are propagated in quadrature, allowing for calculation of corrected** $^{10}$**Be concentrations (see section 3.1). A RSL curve of absolute RSL elevations taken from a GIA model (Bradley et al., 2011) are shown from 8,000 years BP to present day (c).**



## 3.2 The coastal evolution model

Our model combines a rocky profile model (RPM) for rock coast evolution (Matsumoto et al., 2016) with a rock coast, cosmogenic radionuclide production model (Hurst et al., 2017). This coupled model applies a dynamic form of coastal evolution, in which cliff retreat rate is controlled by competing cliff-platform dynamics. Generally, an initial period of rapid
cliff retreat results in widening of the shore platform. As a result, increased wave energy dissipation allows less wave energy to reach the cliff base, and cliff retreat rate declines under stable RSL conditions (Hurst et al., 2017; Trenhaile, 2000; Walkden and Hall, 2005). Either platform lowering or RSL rise can maintain energy supply to the cliffs. As a result, platform morphology is an emergent element of the model.

The exploratory model is conceptualised in a grid discretization framework, in which cells are assigned a binary value of 1 (rock) or 0 (water/air). Wave erosion is considered an erosion driving process and follows established conceptual rocky shore evolution models, which express wave hydraulic and mechanical properties as wave assailing force and considers both horizontal cliff back-wearing and vertical platform lowering (Payo et al., 2015; Sunamura, 1992; Trenhaile, 2008a). Offshore wave height remains fixed throughout a model simulation time; waves are transformed inshore into shallow water and break
when wave height exceeds $0.8 \times$ water depth. Wave height then decays exponentially across the shore platform after wave breaking is initiated. In the model array, each rock cell of the cliff-platform profile is assigned a value for material resistance. The rock cell is eroded and removed from the array (cell values change from 1 to 0) once wave assailing force ($F_w$) exceeds the material resistance value ($F_R$): $F_W \geq F_R$ (Matsumoto et al., 2016).

Subaerial weathering of the platform's intertidal zone acts to lower the resistance of the rock material (Matsumoto et al., 2016). The distribution of intertidal weathering efficacy is informed by empirical experiments of cyclical wetting and drying (Porter et al., 2010). An annual tidal duration distribution (Trenhaile, 2000) is used as an erosion-modulating process by estimating the total annual wave assailing force at each intertidal level (Matsumoto et al., 2016).

Cosmogenic radionuclide production is incorporated into the model by coupling a numerical model of [10]Be accumulation on eroding shore platforms (Hurst et al., 2017). The concentration of [10]Be is calculated for each rock cell at every annual time step. Both [10]Be produced from exposure to neutron spallation at the surface and muon-produced [10]Be at depth are modelled (see section 3.1). Modelling both production pathways for the surface material and at depth below the shore platform surface is important because both horizontal and vertical erosion of the cliff and shore platform are simulated. Horizontal erosion at
the cliff base causes cliff retreat and exposes new shore platform material to spallogenic [10]Be production and accumulation. Concentrations of [10]Be will increase offshore from the cliff base as exposure times increase. Erosion across the intertidal shore platform, including by platform lowering and intertidal weathering, removes the most abundant [10]Be-laden rocks and uncovers rocks with less abundant [10]Be underneath. Incoming cosmic rays are shielded from the shore platform by water





coverage across the platform surface, which is further influenced by tides and RSL change. As water depth increases

offshore, the cosmic ray flux attenuates exponentially and production in the shore platform surface is reduced (Hurst et al., 2017; Regard et al., 2012). Topographic shielding from the presence of a sea cliff also modulates [10]Be production close to the cliff base (Hurst et al., 2017). The combination of scaling factors to account for each of these variables, i.e., production of [10]Be in rock, topographic shielding and water shielding, result in the predicted across-shore 'humped' [10]Be concentration profile (Regard et al., 2012).

## 3.3 Model optimisation

### 3.3.1 Dakota and multi-objective optimisation

We use Sandia National Laboratory's Dakota optimisation software toolkit (Adams et al., 2019) to implement multi-objective optimisation. The optimisation software was chosen to work with the model because of Dakota's flexibility and ease of testing a variety of methods and available functionality within the software. In particular, the Queso Bayesian

calibration library (Estacio-Hiroms et al., 2016) is used to apply the Metropolis-Hastings MCMC algorithm.

### 3.3.2 Objective function definition, scaling and weighting

In this study, we use the coupled model to simulate both a topographic profile and also a [10]Be concentration profile. The first model output is the cliff-platform profile, which displays the elevation, width and gradient of the modelled shore platform in

an across-shore orientation. The second model output is an across-shore [10]Be concentration profile. In order for the model to replicate the topography and [10]Be concentrations of a real, rock coast site, we need to calibrate model results to measured datasets. Our model calibration targets a set of model input parameters that best match the measured data by minimising an objective function (Barnhart et al., 2020). Selected input model free parameters are varied repeatedly within a set parameter space, and model outputs are compared to corresponding data with the aim of minimising residuals between modelled and

measured profiles. Because two outputs are generated with the model, we have two objective functions to minimise simultaneously. Multi-objective optimisation is used to find a set of model input parameters that minimises both topographic and [10]Be concentration residuals combined.

First, the root mean square error (RMSE) is calculated both between the modelled and measured DSM-extracted topographic

profile and also the modelled and measured [10]Be concentration profile, respectively. Modelled outputs and measured data are shifted to the final (present-day) modelled cliff position, where the final cliff position is at 0m. Interpolation is used to assign corresponding modelled data (cell resolution = 0.1 m) to every measured data position across the shore profile. For every measured data point, the elevation and concentration residuals are calculated and combined into a RMSE score for both topographic and [10]Be concentration model outputs. Next, both RMSE values are then scaled ($s_i$) within Dakota to 1) equalise

the magnitude ranges of both the topographic and cosmogenic radionuclide RMSE scores, and 2) set the RMSE magnitudes to a sensible multiple relative to the default measurement error used by Dakota in the likelihood function: variance is



Earth **Surface**
**Dynamics**
Discussions

assumed to be 1.0 when no measurement error is specified. As a result, scaled RMSE scores for both the topographic and [10]Be concentration profiles are within the range of ~0 to 10. Finally, individual weightings ($w_i$) are applied to the scaled RMSE functions for both the topographic and [10]Be concentration profiles (Adams et al., 2019). The scaled and weighted

RMSE scores are combined, and the final composite objective function, *Total RMSE,* becomes:

$$Total\ RMSE\ =\ \sum_{i=1}^{N_o} \frac{\sqrt{w_i}}{s_i}\ \sqrt{\frac{\sum_{j=1}^{N_i}(Mod_{i,j}-Meas_{i,j})^2}{N_i}} \tag{1}$$

In Eq. (1), $N_o$ is the number of individual objective functions we aim to collectively minimise. In this case, we have two

individual objective functions ($N_o = 2$): a topographic profile and a [10]Be concentration profile. Future applications may add additional objective functions ($N_o > 2$), for example, a secondary CRN concentration profile (e.g., [26]Al or [14]C). Weightings applied to the separate RMSE scores are denoted by $w_i$, where subscript $i$ refers to specific values associated with each individual objective function. The weightings applied to the topographic profile and [10]Be concentration profile are changed between MCMC inversion calculations in order to construct the Pareto set of optimised results (see section 3.3.3). The

scaling values are denoted by $s_i$ and are exclusive to the individual objective function. A topographic profile scaling value is calculated by summing the standard error from a linear regression of the topographic profile and the resolution of the UAV imagery. The average measurement error of [10]Be concentrations for each site is used as a scaling value for the [10]Be profile. Table S7 in supplementary materials summarises the objective function scaling values for both sites. For each objective function $i$, the residuals ($Mod_{i,j} – Meas_{i,j}$) are calculated between the modelled and measured data values, which are indexed

by subscript $j$. The number of measured data points are distinct to the topographic profile and [10]Be concentration profile datasets and are denoted by $N_i$.

### 3.3.3 Pareto front results

When performing multi-objective optimisation, rather than a single optimal solution, there are multiple optimised solutions,

which map out what is known as a Pareto front. We need to consider best-fit model results across a spectrum of objective function combinations, because changing the weightings applied to each objective function may result in different best-fit input model parameters. The Pareto front is a set of optimised results for which no improvement can be made to an individual objective function without compromising the performance of at least one of the other objective functions. This set of results is the most optimal set of input model parameters. The Pareto front is constructed by performing numerous MCMC

inversions with various weightings given to the RMSE scores that are calculated for the topographic and [10]Be concentration profiles for each run. The weighted RMSE values are summed to form a single objective function that Dakota aims to minimise (Eq. 1). In this investigation, a total of five MCMC calculations for each site are performed. For each of the five MCMC runs, the weightings assigned to the topographic and [10]Be concentration profile RMSE scores are changed.



Weightings assigned to each individual objective functions for each MCMC analysis are shown in Table 1. Figure 3 shows a

basic framework of the multi-objective optimisation of the coupled model.

**Table 1: Weightings assigned to the topographic and $^{10}$Be concentration RMSE scores for the five MCMC calculations.**

| MCMC analysis | Topography weighting (%) | $^{10}$Be weighting (%) |
|---|---|---|
| 1 | 50 | 50 |
| 2 | 25 | 75 |
| 3 | 75 | 25 |
| 4 | 5 | 95 |
| 5 | 95 | 5 |










**Figure 3: Structure for implementing a single MCMC calculation using Dakota. Data inputs into the coupled model include a topographic profile, a $^{10}$Be concentration profile and a RSL history. The MCMC analysis is performed multiple times with different weightings for the objective functions (topographic profile RMSE and $^{10}$Be concentration profile RMSE) and produces a corresponding maximum likelihood estimation (MLE\*) result. For each MCMC calculation, the Weights\* value is changed for each RMSE score. The different values for the Weights\* are shown in Table 1 and correspond to $w_i$ (Eq. 1). The set of MLE results together produce the 'Pareto front' of multi-objective optimised results.**



### 3.3.4 MCMC analysis

Metropolis-Hastings is a specific MCMC implementation (Metropolis et al., 1953), in which MCMC is a class of methods based on Bayesian inference calibration. A detailed explanation of how Bayesian inference can be used to calibrate models is provided by Kennedy and O'Hagen (2001).

The composite RMSE score (Eq. 1) is calculated and input into a Gaussian likelihood function in Dakota; the lowest RMSE

score results in the maximum likelihood estimation (MLE). A so-called proposal distribution is used to select and jump to new parameter values within the MCMC algorithm. After each run, new values for the free parameters $y$, $F_R$ and $K$ (see section 3.4) are randomly selected from a uniform proposal distribution centred at the current accepted parameter values. A likelihood ratio compares the posterior likelihood of the proposed parameter set to the previous accepted likelihood and is used to decide whether the new set of parameters is accepted or rejected. If the proposed parameter set produces a model

result that is more likely than the current accepted parameter set (ratio of current to last accepted iteration >1), then the new parameter set is always accepted. If the proposed posterior is less likely (likelihood ratio <1), the new parameter set may still be accepted with a probability of acceptance proportional the likelihood ratio (Hurst et al., 2016). This is achieved by generating a random number $r$ from a uniform distribution between 0 and 1; if $r <$ ratio, the proposed parameter set is accepted. The Metropolis-Hasting algorithm allows for acceptance of less likely parameter sets in order to prevent the

acceptance chain from reaching an immovable position in a localised likelihood trough. The proposal distribution variance was tailored so as to produce an acceptance rate of ~23% that ensures optimal chain mixing and full exploration of the parameter space (Gelman et al., 1997).

As we have no prior knowledge of the best-fit model parameters, a uniform prior distribution is used. As the prior

distribution is essentially removed from the posterior probability calculation, Dakota returns best-fit parameter values that correspond with the MLE, which is similar to methods used to find optimal model results by Hurst et al. (2016). Dakota takes the log form of the likelihood function to help numerical stability by working with more manageable negative numbers and transforming from multiplications to additions. Minimising the negative log-likelihood is equivalent to maximising the likelihood.


### 3.3.5 Dakota functionality and constraining the parameter space

The parameter space is constrained to where modelled topographic profiles are similar to those observed at the selected study sites. The failure capture recovery option within Dakota is used to identify which combinations of input model free parameters cannot replicate the measured topographic profile sufficiently. A 'fail' flag is produced by the model if the

modelled shore platform profile does not erode to a width of at least the intertidal width of the measured shore platform profile. The width of the measured topographic profile (~250 m) should be taken as the minimum width because the shore platform undoubtedly extends further offshore than where the UAV survey ended. A 'fail' flag returned to Dakota is



replaced by a high RMSE value (set to 999999) so this combination of input model free parameters is avoided in future simulations within the Metropolis-Hastings algorithm.


### 3.4 MCMC analysis inputs

A previous investigation into the relative importance of wave erosion versus weathering using the RPM coastal evolution model found that wave erodibility, material resistance and weathering rate parameters have the greatest influence on the dominance of erosional form (Matsumoto et al., 2018). Because these model variables have the greatest control over

principal erosion processes, wave erodibility by means of wave height decay rate ($y$), material resistance ($F_R$) and maximum intertidal weathering rate ($K$) are chosen to vary in the MCMC calculations. In order to replicate a full range of platform geometries, we vary these parameters over several orders of magnitude across a range that was guided by previous exploratory morphodynamic modelling (Matsumoto et al., 2016, 2018). We need to not only apply effective proposal distributions to the free parameters that ensure the full parameter space is explored, but also achieve optimal acceptance rates

(see section 3.3.4). In order to target optimal acceptance rates and fully explore the parameter space, exponents of the $y$, $F_R$ and $K$ variables are treated as the calibration parameters, similar to approaches taken by Barnhart et al. (2019) and these values are varied between model runs. Symbology assigned to the exponent calibration parameters are $a$ for $y$, $b$ for $F_R$ and $c$ for $K$, which are summarised in Table 2. As these parameters are abstractly defined within the model, it is important to highlight that our aim is not to report accurate wave force, weathering rates and material strength at real-world sites, but to

determine the best combination of model parameters to match measured datasets in order to model cliff retreat.

Wave erodibility is explored in the MCMC analysis by varying the wave height decay rate ($y$), which is consistent with previous modelling approaches (e.g., Matsumoto et al., 2018; Trenhaile, 2000). Erosion achieved by breaking and broken waves can be changed by varying the distance across the shore platform that waves can dissipate energy: wave height decay

rate ($y$). A small value for $y$ means wave height will decay slowly, in which case breaking waves exert energy across a greater distance of the shore platform surface, which achieves more erosion. In contrast, a large value for $y$ indicates that wave height will decay quickly and wave-driven erosion covers a shorter distance across the shore platform. Initial ranges of $y$ followed Matsumoto et al. (2018) by varying $a$ between -2 and 1 (i.e. $y = 0.01\text{-}10$ m$^{-1}$) which, across a 200 m-wide shore platform, would equate to wave height decrease by 67-100%. These wave attenuation rates are consistent with field-based

observations of wave transformation across a shore platform. For example, Ogawa et al. (2011) finds wave height was reduced by ~93% across a 250 m-wide platform at the lowest tidal stage. Wave height decay rate $(y)$ is further constrained in this study to 0.01-0.16 m$^{-1}$, as when values of $y$ exceed 0.16 m$^{-1}$, wave height would dissipate too quickly and wave force is not large enough to erode a shore platform to a distance that matches at least the width of the measured intertidal platform for both the Bideford and Scalby sites. This results in failed model runs as detected by the failure capture recovery function

in Dakota (see section 3.3.5).





For material resistance, we use a range of $b$ from 1-3 (i.e. $F_R$ = 10-1000 kg m$^2$ yr$^{-1}$), which follows Matsumoto et al. (2018), and encompasses material resistance values used by other modelling studies that explore a range of rock strength settings, e.g., Trenhaile (2008a, b, 2000). The conceptual value for material resistance ($F_R$) is highly simplified by incorporating
mechanical, geological and structural rock factors into a single value to represent rock mass strength (Matsumoto et al., 2016).

Maximum intertidal weathering rate ($K$) is varied as a proportion of the material resistance (Matsumoto et al., 2018). Maximum weathering rate occurs at the mean high water neap tidal level (MHWN), which is defined by a weathering
efficacy distribution (Porter et al., 2010). The greatest rate of weathering that we apply is equal to: $F_R \times 0.2$ kg m$^2$ yr$^{-1}$, which, results in maximum down-wearing rates of 20 mm yr$^{-1}$ when only considering weathering contributions to shore platform downwear. In this study, preliminary investigations were carried out to establish an appropriate range of weathering rates needed to match both objective functions at the two UK sites. Initial results for Bideford and Scalby showed that the topographic profiles and $^{10}$Be concentration profiles could only be well-matched simultaneously at very low to negligible
weathering rates ($K$). Weathering of the shore platform surface becomes negligible when weathering rates ($K$) fall below a particular ratio in relation to material resistance ($F_R$). By calculating the weathering rate ($K$) (kg m$^{-2}$ y$^{-1}$) for the full modelled simulation time ($K$ x 8,000 years), we can compare this to the value of $F_R$ to detect when weathering rate ($K$) is less than the material resistance ($F_R$). We find that when the exponent of $c < -5$, maximum weathering rate falls below the value for material resistance ($F_R$) for the simulated model time, and erosion of the shore platform through weathering processes
becomes negligible, because rock cells cannot be removed from the model array by means of subaerial weathering. The range of $c$ was adjusted to -10 to -4 for Bideford and -10 to -1 at Scalby to ensure negligible weathering was included within the MCMC analysis parameter space. At Scalby, $^{10}$Be concentrations can still be matched at higher rates of weathering, therefore we include the upper range of $K$ values in the MCMC analysis. The wide range of weathering rates that we explore are similar to equivalent platform downwear rates quantified for a range of field-based studies (e.g., Buchanan et al., 2020;
Moses, 2014; Stephenson et al., 2019; Swirad et al., 2019).

Other fixed model parameters are set to the same values as used by Matsumoto et al. (2018) (supplementary materials Table S7). The RSL history input is taken from the GIA model of Bradley et al. (2011). A fixed mean spring tidal range of 8.41 m for Bideford and 4.6 m for Scalby are used, which are based on tide gauge records (National Tidal and Sea Level Facility,
2021). Table 2 summarises the ranges of the free parameters $y$, $F_R$ and $K$ used in the MCMC analysis.







**Table 2: Free variables, corresponding calibration parameters and their units, and lower and upper bounds used in the MCMC calculations.**

| Variable description | Calibration parameter (exponent) | Upper and lower bounds $(a,b,c)$ | Variable | Equivalent variable range $(y, F_R, K)$ | Units |
|---|---|---|---|---|---|
| Wave height decay rate | $a$ | -2 ____ -0.8 | $y$ $(= 10^a)$ | 0.01-0.16 | $(m^{-1})$ |
| Material resistance | $b$ | 1 ____ 3 | $F_R$ $(=10^b)$ | 10-1000 | $(kg\ m^{-2}\ yr^{-1})$ |
| Maximum weathering efficacy rate | $c$ | -10 ____ Scalby: -1 Bideford: -4 | $K$ $(=5^c \times F_R)$ | Scalby: $10^{-6}$-200 ____ Bideford: $10^{-6}$-1.6 | $(kg\ m^{-2}\ yr^{-1})$ |

## 3.5 Interpretation of results

The best-fit parameter results provided by Dakota, which correspond to the set of parameters with the MLE, are used for the model results that best-fit the measured data. Likelihood-weighted histograms are constructed from the distribution of

accepted MCMC samples for each of the free parameters. Confidence intervals defined by the 16% and 84% percentiles of the distributions are used as the uncertainty for the best-fit parameter values. We choose not to use 5% and 95% confidence intervals because the resultant range of model outputs produce unrealistic uncertainty for the modelled topographic profile in terms of the range in platform elevations and gradients. To observe the resultant uncertainty of model outputs as defined by the MCMC results, ensemble runs of the coupled model explore the median, 16 and 84% confidence range for each

parameter against the median result of the other two parameters. A total of nine model outputs for each site are produced.

## 4 Results

### 4.1 Pareto set of optimised results

A Pareto set was constructed by performing multiple MCMC inversions with different weightings given to the two objective

functions (see section 3.3.3). Because we use measurement error to scale each objective function, there is potential for bias as a result of the relative precision of both measured datasets. It was necessary to explore a range of weightings for the objective functions to understand the impact of measurement error to the model outputs and to explore how sensitive final, best-fit model outputs were to changes in the dominant objective function.



The Pareto set of MCMC best-fit results for the Bideford site is shown in Figure 4a. All combinations of objective function weightings, except for the MCMC analysis weighted most towards the $^{10}$Be concentration profile (shown in darkest red), fit both measured datasets well. The MCMC best-fit result with 95% weighting assigned to the $^{10}$Be concentration profile and 5% assigned to the topographic profile results in a topographic profile that is considerably steeper than the measured profile at Bideford. We do not want to base modelled cliff retreat rates on scenarios that are not able to replicate well the topography

of the shore platform profile, and so should avoid weighting too strongly towards the $^{10}$Be concentration profile. Dissimilarly, at Scalby, all combinations of objective function weightings produce very similar model outputs (Fig. 4b). This reveals that the best-fit model result for the topographic profile and the $^{10}$Be concentration profile are found in the same parameter space for Scalby, but not necessarily for Bideford. Crucially, final results from the 50 – 50% weighted MCMC analysis show a good representation of the full Pareto set of result (Fig. 4). We suggest that future applications can

confidently use a single, equally-weighted multi-objective MCMC calculation to optimise the coupled model to multiple measured datasets and quantify modelled cliff retreat rates. Subsequently, results from the equally-weighted MCMC analysis are used to construct final model outputs and objective function surfaces in following sections. Results from all weighted MCMC calculations can be found in the supplementary materials, Table S8.



Earth **Surface**
**Dynamics**
Discussions

EGU




**Figure 4: The five Pareto set results for both Bideford (a) and Scalby (b) sites. The modelled topographic profile and $^{10}$Be concentration profile are shown alongside corresponding measured data. Modelled cliff retreat rates are shown for the past 7000 years. Grey coloured model results correspond to 50 – 50% objective function weighted MCMC results. Darkest blue coloured model results correspond to the MCMC results that were most weighted towards the topographic (Topo) profile (95%). Darkest red coloured model results correspond to the MCMC results that were most weighted towards the $^{10}$Be concentration (CRN) profile (95%).**





## 4.2 Model results from multi-objective optimisation

Model outputs using the best-fit results and uncertainty defined by the 50 – 50% weighted MCMC results show that the
coupled model is able to produce a good fit to both the topographic profile and $^{10}$Be concentration profile for both sites (Fig.
5). For the topographic profile, maximum uncertainty in the elevation range is found furthest offshore from the cliff. For the
intertidal platform width, where measured data was collected, maximum uncertainty in elevation is ~5 m (300 m offshore
from the cliff) at Bideford (Fig. 5b) and ~1 m (300 m offshore from the cliff) at Scalby (Fig. 5f). Modelled $^{10}$Be
concentrations display the characteristic 'humped' profile (Regard et al., 2012) (Fig. 5c and Fig. 5g), with maximum
variance in $^{10}$Be concentrations occurring offshore of the 'hump' at both sites. Maximum uncertainty in modelled $^{10}$Be
concentrations for the intertidal platform is ~ 5000 atoms g$^{-1}$ at Bideford (Fig. 5d) and ~2500 atoms g$^{-1}$ at Scalby (Fig. 5h).
These uncertainties are proportional to the magnitude of the measured concentrations, with peak $^{10}$Be concentration of 14818
atoms g$^{-1}$ at Bideford and 7547 atoms g$^{-1}$ at Scalby.

At Bideford, the best-fit modelled scenarios show the platform has eroded to a width of 750-1650 m across the past 8000
years (Fig. 5a). At Scalby, however, the best-fit modelled scenarios indicate the platform has eroded to a wider width of
2200-3500 m (Fig. 5e), over the same time period. Time stamps for modelled cliff positions are back calculated and shown
for the corresponding distance across the shore platform. For example, the modelled cliff position at Bideford was 200 m
offshore from the present-day cliff position ~5000 yrs BP (Fig. 5b). Our results indicate that the 250m, present-day intertidal
platforms at Bideford and Scalby were eroded in the past 5800 and 2400 years, respectively (Fig. 5). The lesser time taken to
erode the same distance of shore platform at Scalby indicate that late-Holocene cliff retreat rates at Scalby are over two times
faster than at Bideford. As the magnitude of the measured $^{10}$Be concentrations at Scalby is considerably less than at
Bideford, this model result aligns with model predictions of CRN concentrations and cliff retreat rate interpretations, i.e.,
that the magnitude of CRN concentrations is inversely proportional to cliff retreat rates.

Earth **Surface**
**Dynamics**
Discussions
EGU

**Figure 5: Final results from the 50 - 50 % weighted multi-objective MCMC calculation for the Bideford study site (a,b,c,d; shown in purple) and Scalby study site (e,f,g,h; shown in orange). Dark lines show the best-fit results and the shaded areas show the confidence interval uncertainty range. The 16% - 84% confidence interval for each parameter ($F_R$, $K$ and $y$) were plot against the median results for the other parameters and shaded uncertainty regions were constructed from the upper and lower limits of model outputs. Subplots (a) for Bideford and (e) for Scalby show the full width of the modelled topographic profile. Subplots (b) for Bideford and (f) for Scalby show the first 300 m of the modelled platform offshore from the cliff (0 m) and compares modelled results to the measured topographic profile (black solid line). Subplots (c) for Bideford and (g) for Scalby show the full modelled extent of the $^{10}$Be concentration profile. Subplots (d) for Bideford and (h) for Scalby, show the first 300 m of the modelled $^{10}$Be concentration profile offshore from the cliff (0 m) and compares modelled results to the measured and corrected, $^{10}$Be concentrations (see section 3.1). Timestamps for the cliff positions across the 8000-year duration simulation time were back calculated and shown against corresponding cliff positions.**





A time series of cliff retreat rates, which are calculated from modelled cliff positions every 100 years, show a decline in cliff retreat rates across the Holocene (Fig. 6). The evolution of modelled cliff retreat covered by the across-shore distance of measured data encompasses the past 5800 years at Bideford (Fig. 5b) and the past 2400 years at Scalby (Fig. 5f). We are only able to report cliff retreat rates with confidence for these time periods which correspond to the measured datasets. Following this, the model results for Bideford show cliff retreat rates have declined from 7.5-17.5 cm yr$^{-1}$ at 5800 yrs BP to 1-3 cm yr$^{-1}$ at present day (Fig. 6a). Likewise, the model results for Scalby show cliff retreat rates have declined from 11-17 cm yr$^{-1}$ at 2400 yrs BP to 4-8 cm yr$^{-1}$ at present day (Fig. 6b). Despite lacking measured data beyond 250 m offshore from the cliff, we can assess the antecedent modelled cliff retreat rates needed to match the modelled to measured profiles in the intertidal zone (full extent of grey areas in Figure 6). Best-fit model results for the full model simulation time reveal a declining cliff retreat rate scenario for both sites. At Bideford, cliff retreat rates decline from 25-55 cm yr$^{-1}$ at 7000 yrs BP to 1-3 cm yr$^{-1}$ at present day (Fig. 6a). Similarly, at Scalby, cliff retreat rates decline from 70-100 cm yr$^{-1}$ at 7000 yrs BP to 4-9 cm yr$^{-1}$ at present day (Fig. 6b).

The best-fit scenario of a declining cliff retreat rate throughout the Holocene directly reflects the pattern of decline in the rate of RSL rise for both UK sites. At Bideford, the rate of RSL rise falls from +0.25 mm yr$^{-1}$ at 5800 yrs BP to +0.05 mm yr$^{-1}$ at present day (Fig. 6a). Similarly, at Scalby the rate of RSL rise falls from +0.11 mm yr$^{-1}$ at 2700 yrs BP to +0.03 mm yr$^{-1}$ at present day (Fig. 6b).

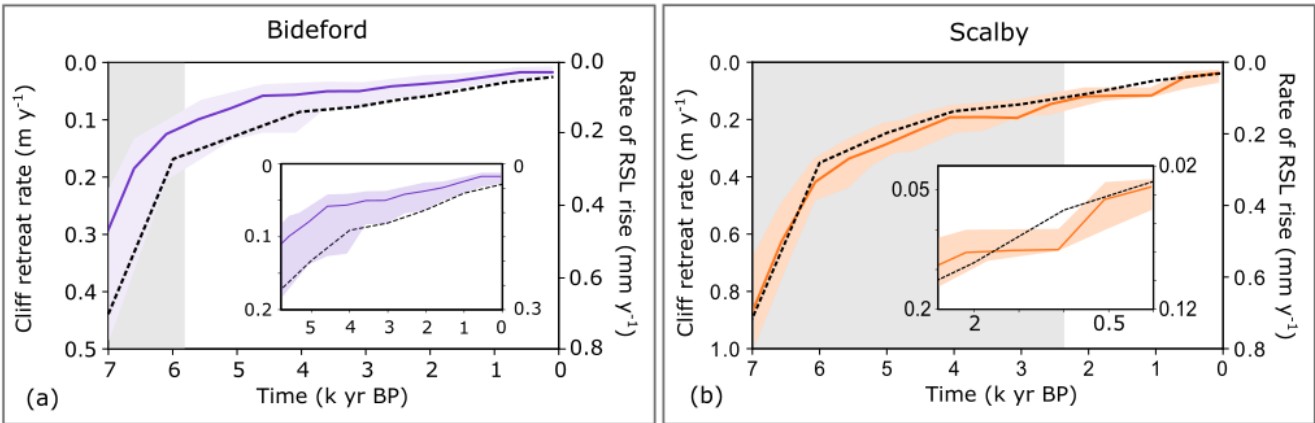

**Figure 6: A time series of cliff retreat rate (m y$^{-1}$) shown by the solid line and shaded area across the late Holocene (from 7000 yr BP to present day) calculated from modelled cliff positions every 100 years. The rate of RSL rise (mm y$^{-1}$) is shown alongside cliff retreat rates by the dashed black line. Inset plots show the cliff retreat rates and rate of RSL rise for the time period that correspond with the distance across the shore platform over which measured data were analysed (~250 m). For Bideford a), the past 5800 years of modelled cliff retreat correspond with the 250 m intertidal shore platform. For Scalby (6b), the past 2700 years of modelled cliff retreat correspond with the 250 m intertidal shore platform. The grey shaded area highlights the projected cliff retreat rates based on modelled results, but do not directly correspond with time period associated with the measured data.**





Table 3 summarises the results from the 10,000 iteration, equally-weighted MCMC analysis for both sites, with uncertainties defined by the 16% and 84% confidence intervals of the likelihood-weighted accepted sample distributions. Acceptance rates

for all MCMC calculations ranged between 17% and 40%, and therefore encompass the range of optimum acceptance rates for chain mixing (Gelman et al., 1997). Supplementary Figure S1 and Figure S2 plot the cumulative moving median, 16% and 84% quantiles across the 10,000 iterations to show the MCMC burn-in period. We found that 10,000 iterations for each weighted MCMC analysis was a sufficient number of samples to build robust posterior distributions.

**Table 3: Best-fit parameter results from 50 – 50% weighted, multi-objective MCMC calculations.**

| Wave height decay rate ($y$) | | | Material resistance ($F_R$) | | Weathering rate ($K$) | |
|---|---|---|---|---|---|---|
| **Bideford** | | | | | | |
| $a$ | $y$ | | $b$ | $F_R$ | $c$ | $K$ |
| -1.45 | $\begin{array}{c} +0.09 \\ \hline -0.24 \end{array}$ | 0.02-0.04 m$^{-1}$ | 1.93 | $\begin{array}{c} +0.68 \\ \hline -0.56 \end{array}$ 23-407 kg m$^{-2}$ yr$^{-1}$ | -6.14 $\begin{array}{c} +0.17 \\ \hline -2.83 \end{array}$ | ~$10^{-4}$-$10^{-3}$ kg m$^{-2}$ yr$^{-1}$ |
| **Scalby** | | | | | | |
| $a$ | $y$ | | $b$ | $F_R$ | $c$ | $K$ |
| -1.97 | $\begin{array}{c} +0.17 \\ \hline -0.02 \end{array}$ | 0.01-0.015 m$^{-1}$ | 2.04 | $\begin{array}{c} +0.1 \\ \hline -0.76 \end{array}$ 19-138 kg m$^{-2}$ yr$^{-1}$ | -4.94 $\begin{array}{c} +1.08 \\ \hline -3.61 \end{array}$ | ~$10^{-4}$-0.2 kg m$^{-2}$ yr$^{-1}$ |

### 4.2.1 Material resistance ($F_R$)

Posterior MCMC results of accepted samples show that the topography and $^{10}$Be concentration profiles can be well-matched across a large range of $F_R$ values (Table 3). For Bideford, the best-fit result defined by the 16-84% confidence intervals for the equally weighted multi-objective MCMC analysis for $F_R$ is equivalent to 23-407 kg m$^{-2}$ yr$^{-1}$. For Scalby, best-fit results for $F_R$ tend towards lower values for material resistance of 19-138 kg m$^{-2}$ yr$^{-1}$. The large uncertainty for $F_R$ from posterior distributions is caused by correlation found between $F_R$ and $y$ (see section 5.2).


### 4.2.2 Weathering rates ($K$)

At both sites, low to negligible weathering rates ($K$) as a proportion of material resistance ($F_R$) are needed to match both measured datasets simultaneously. At Bideford, the best-fit results for maximum weathering rate are skewed towards the lowest $K$ values. Both measured datasets could only be well-matched when negligible weathering was implemented in the

modelled profile evolution. Best-fit results for $K$ range across orders of magnitudes (~$10^{-5}$-$10^{-3}$ kg m$^{-2}$ yr$^{-1}$) when applying





the best-fit result for $F_R$ (85 kg m$^{-2}$ yr$^{-1}$). Using the upper limit of $F_R$ and $K$ defined by 84% confidence intervals, the absolute maximum weathering rate acting on the profile at Bideford is equivalent to 0.03 kg m$^{-2}$ yr$^{-1}$. This weathering rate would result in maximum downwear rates equivalent to 0.007 mm yr$^{-1}$ when only considering downwear as a result of intertidal platform weathering. These results indicate that erosion of the shore platform at Bideford is dominated by wave-driven erosion.

In comparison to Bideford, at Scalby, maximum weathering rates acting on the shore platform surface ($K$) best match the measured data at intermediate to low weathering rates, equivalent to ~ 0.0001-0.27 kg m$^{-2}$ yr$^{-1}$ when using the best-fit result of $F_R$ (109 kg m$^{-2}$ yr$^{-1}$). This weathering rate would result in maximum downwear rates equivalent to 0.2 mm yr$^{-1}$ if only considering downwear as a result of intertidal platform weathering. Although weathering rates are still low at Scalby, distinctions between the two sites in relation to maximum weathering rates can still be made. Observing objective function surfaces (see section 4.3) can help explain the differences between the two UK sites in terms of weathering-controlled erosion of the shore platform and the match to both the measured topography and [10]Be concentrations. This, in turn, can reveal if and where any compromises had to be made to match both objective functions, e.g., if the two individual objective functions are not minimised in the same area of the parameter space.

### 4.2.3 Wave height decay rates ($y$)

Results from the wave height decay rate ($y$) show Scalby best-fit results tend towards the lowest wave height decay rates, which are equivalent to 0.01-0.015 m$^{-1}$. Or, in other words, breaking waves exert erosive power across the furthest distance possible, based on realistic constraints placed on $y$ (see section 3.4). Bideford on the other hand, has best-fit $y$ values at higher wave height decay rates, with a best-fit value equivalent to 0.02-0.04 m$^{-1}$. The larger value for $y$ means wave height will dissipate faster and cover a shorter distance across the shore platform at Bideford in comparison to Scalby.

### 4.3 Objective function results

Observing the objective function space for the topographic RMSE, [10]Be concentration RMSE and the combined likelihood helps disclose the trade-offs between the three different parameters ($Fr$, $K$ and $y$) and the effects on the two objective functions. In following sections, when addressing the unitless values for the free parameters, we are referring to the exponential calibration parameters ($a,b,c$) that were varied in the MCMC calculations (see section 3.4).

### 4.3.1 Material resistance vs. weathering rate

Comparisons between calibration parameters varied for material resistance ($b$) and weathering rate ($c$) show no distinctive influence on the endmember topographic profile at the Bideford site (Fig. 7a). In contrast, only once $c$ falls below -5 and weathering rates become negligible, can the modelled [10]Be concentration profile closely replicate the measured data (Fig. 7b). Furthermore, with insignificant active weathering, combinations between material resistance and weathering rate show





no noteworthy influence on the modelled $^{10}$Be concentration profile. The likelihood objective function for the accepted samples show the best samples are generally found when $c$ is below -5 (Fig. 7c). Results from initial Bideford MCMC calculations, where the range of $c$ is set to -5 to -1 are found in the supplementary material and show that the $^{10}$Be concentrations cannot be matched with the higher range of weathering rates explored initially (Figure S3 supplementary materials). There needs to be negligible intertidal weathering of the shore platform in order to produce the relatively high

$^{10}$Be concentrations measured at the Bideford site. This is only revealed when optimising for the $^{10}$Be profile; optimising for the topographic profile alone, however, does not reveal any distinct behavioural trends between $b$ and $c$.

At Scalby, comparisons between $b$ and $c$ show a better fit to the topographic profile, can generally be found at the lower range of $b$ and $c$ (Fig. 7d). In contrast, the $^{10}$Be concentration profile can be well-matched at higher weathering rates, where $c$

is -2 (Fig. 7e). These results contrasts with the results from Bideford, which show negligible weathering needs to occur in order to match the $^{10}$Be concentration profile. However, these relationships are not well-defined: well-matched $^{10}$Be concentration profiles can still be produced when weathering rates are low (Fig. 7e). Equally, poorly-matched topographic profiles can still be produced at the lower range of $b$ and $c$ (Fig. 7d). This suggests that the 3$^{rd}$ free parameter varied in the MCMC analysis, wave height decay rate ($a$), has further influence on the final modelled profiles.

Earth **Surface**
**Dynamics**
Discussions


**Figure 7: Objective function results for the material resistance (*b*) and weathering rate (*c*) parameters; constructed from 50 – 50% weighted MCMC analysis results for the Bideford site and Scalby site. The topographic profile RMSE [(a) and (d)] and $^{10}$Be concentration profile RMSE [(b) and (e)] objective function spaces are constructed from all 10,000 samples visited in the MCMC analysis. The combined likelihood objective function space [(c) and (f)] is constructed just from the accepted samples from the MCMC analysis. Dark blue corresponds with the worst samples that have the highest RMSE and negative log-likelihood scores, while bright yellow corresponds with the best samples that have the lowest RMSE and negative log-likelihood scores.**







### 4.3.2 Wave height decay rate vs. weathering rate

For Bideford, a clear zone in the mid-range of wave height decay rates ($a$ = -1.5) produces well-matched topographic profiles across the full range of weathering rates ($c$) (Fig. 8a). When $a > -1.5$, a threshold exists where a zone of well-matched topographic profiles suddenly meets a zone of poorly matched topographic profiles. So, when wave height decay rate ($a$) is increased too much, waves will dissipate their energy too quickly and will result in modelled topographic profiles with gradients too steep to match the topographic profile at Bideford.

When considering the influence on the [10]Be concentration profile at Bideford, a much wider zone across the range of $a$ produces well-matched [10]Be concentration profiles when $c < -5$, i.e., when there is negligible subaerial weathering occurring (Fig. 8b). This zone of well-matched [10]Be concentration profiles extends across the boundary where good and bad topographic profiles can be produced (Fig 8a). This results in a dense zone of accepted samples across a range for $a$ of ~ -1.8 to -1.3 when $c < -5$ (Fig. 8c).

In contrast to Bideford, the slowest wave height dissipation across the shore platform (defined by the lowest $a$ values) produces the best topographic profile seen at Scalby (Fig. 8d). Only when assessing the [10]Be concentration profile are we able to see how weathering rates interact with wave erosion to impact the model results for Scalby (Fig. 8e). Slower wave height dissipation requires lower rates of weathering in order to produce a matching [10]Be concentration profile. In the zone of well-matched results, as wave height decay rate ($a$) values decrease, weathering rates also decrease (Fig. 8e). Because the peak in [10]Be concentrations at Scalby is ~7000 atoms g[-1] less than the peak in [10]Be concentrations at Bideford, greater weathering rates, and as a result, greater platform lowering can produce a modelled [10]Be profile that matches the measured data for Scalby. As active subaerial weathering can contribute to model outputs that can replicate the measured data, the balance of wave driven and weathering controlled erosion is more complex at Scalby. Wave erosion and weathering can trade-off in multiple combinations to produce model outputs that match both the measured [10]Be concentration profiles and topographic profiles at Scalby. The accepted samples (Fig. 8f) show that the best model results are found at the lower range of $y$, when constrained by matching to the topographic profile (Fig. 8d) and [10]Be profile (Fig. 8e) simultaneously.

Earth **Surface**
**Dynamics**
Discussions



**Figure 8: Objective function results for the wave height decay rate (y) and weathering rate (K) parameters constructed from 50 – 50% weighted MCMC analysis results for the Bideford and Scalby sites. The topographic profile RMSE [(a) and (d)] and $^{10}$Be concentration profile RMSE [(b) and (e)] objective function spaces are constructed from all 10,000 samples visited in the MCMC analysis for Bideford and Scalby, respectively. The combined likelihood objective function space [(c) and (f)] is constructed just from the accepted samples from the MCMC analysis. Dark blue corresponds with the worst samples that have the highest RMSE and negative log-likelihood scores, while bright yellow corresponds with the best samples that have the lowest RMSE and negative log-likelihood scores.**





### 4.3.3 Wave height decay rate vs. material resistance

A distinct, negative relationship exists between material resistance ($b$) and wave height decay weight ($a$) at Bideford when optimising to the topographic profile (Fig. 9a). As $b$ increases, more wave energy is needed to erode the cliff at a rate fast

enough to produce a shore platform with a gradient shallow enough to match that seen at Bideford. In other words, wave height decay rate has to reduce so waves can dissipate their energy across a further distance. Varying the wave decay rate ($a$) for any value of $b$ reveals a distinct zone where wave energy is high enough to produce a wide enough platform, but not large enough to decrease the profile slope to below the observed 0.02 gradient at Bideford. The same relationship exists when optimising for the [10]Be concentration profile, but the best region has shifted to a zone with a greater wave height decay

rate (Fig. 9b). This ratio between $a$ and $b$ strongly controls the accepted results for Bideford (Fig. 9c). Section 5.2 expands on this correlation found between the $a$ and $b$ parameters.

For Scalby, there is a similar, but less distinctive, trade-off between $a$ and $b$ as seen at the Bideford site. A wider range of wave height decay rates ($a$) is able to produce a suitable topographic profile in relation to $b$; with the most likely results

tending towards lower rates of wave height decay rate ($a$) (Fig. 9d). The platform gradient is shallower at Scalby (0.01); once $a$ falls below ~ -1.5, a shallow platform gradient can be produced by a wider range of wave height decay rate values, as long as material resistance ($b$) is relatively low. The influence of the [10]Be concentration optimisation constrains the window of best results (Fig. 9e). Accepted samples cover a wider area of the parameter space at Scalby compared to Bideford (Fig. 9f).

Earth **Surface**
**Dynamics**
Discussions




**Figure 9: Objective function results for the material resistance (*b*) wave height decay rate (*a*) parameters; constructed from 50 –
50% weighted MCMC analysis results for the Bideford site and Scalby site. The topographic profile RMSE [(a) and (d)] and $^{10}$Be
concentration profile RMSE [(b) and (e)] objective function spaces are constructed from all 10,000 samples visited in the MCMC
analysis. The combined likelihood objective function space [(c) and (f)] is constructed just from the accepted samples from the
MCMC analysis. Dark blue corresponds with the worst samples that have the highest RMSE and negative log-likelihood scores,
while bright yellow corresponds with the best samples that have the lowest RMSE and negative log-likelihood scores.**






## 5 Discussion

Our ultimate aim is to quantify the long-term history of transient cliff retreat rates in order to enable better predictions of future cliff retreat rates at rock coast sites across the UK and world-wide. Our results show that rigorous multi-objective optimisation of a process-based coastal evolution model to high-precision measured datasets, permits long-term trajectories

of cliff retreat to be identified and quantified for real-world sites over centennial to millennial timescales.

To explore the potential for further application of these methods at other rock coast sites, here we justify the methodology chosen, address equifinality in model results and explore how equifinality impacts our ability to quantify cliff retreat rates. We also highlight some aspects of our results that should be interpreted with caution, specifically where correlation exists

between parameters.

### 5.1 The importance of multi-objective optimisation

Dissimilar patterns in the objective function space between the topographic profile RMSE and the $^{10}$Be concentration RMSE are revealed in Figures 7, 8 and 9. We, therefore, do not have the confidence to rely solely on one objective function to

accurately model the evolution of real-world rock coasts. Using the results from Bideford as an example, we can observe model outputs for three different zones within the parameter space: 1) zones where we are able to match the topographic profile, but not the $^{10}$Be concentration profile; 2) zones where we are able to match the $^{10}$Be concentration profile, but not the topographic profile; and 3) zones where we are able to match both objective functions.

At the Bideford site, the parameter space that can produce model results that match the topographic profile well, but the $^{10}$Be concentration profile poorly, is only found where weathering rates are high ($c > -5$) (shaded blue, Fig. 10a). Examples of model outputs from this area (Fig. 10a) of the parameter space show that the shore platform gradient and elevation can be matched well, but the magnitude of all modelled $^{10}$Be concentration profiles are considerably lower than the corresponding measured data (Fig. 10b). This upper range of weathering rates ($c$) was incorporated into the initial MCMC analysis for the

Bideford site, but was adjusted in later simulations because the model could not produce well-matched $^{10}$Be concentration profiles for this range of $c$ (see section 3.4). If we only optimise to the modelled topographic profile, we could vastly overestimate the contributions of weathering to the shore profile evolution at Bideford. More importantly, modelled cliff retreat rates are consistently faster than the multi-objective optimised retreat rate results that also try to match $^{10}$Be concentrations (Fig. 10e).


Furthermore, the zone where model outputs can match the $^{10}$Be concentration profile, but cannot match the topographic profile is shown for the parameter space between material resistance ($b$) and wave height decay rate ($a$) (shaded orange, Fig. 10a). In all corresponding model examples shown (Fig 10c), the magnitude of the $^{10}$Be concentration profile matches the





measured data well, but modelled shore platform profiles are steeper than the measured topographic profile. As wave height
decay rate values (*a*) are increased above the parameter space that is able to match both objective functions well (shaded
pink, Fig10a), the reduction in wave erodibility produces steeper profiles as erosion is less efficient. Concentrations of $^{10}$Be
are highly dependent on the evolution of the surface topography. So, even if we are able to match $^{10}$Be concentrations to
corresponding data, if the topography is incorrect, we are matching the $^{10}$Be concentrations incidentally and these results
should be discarded.

Finally, the parameter space where both the topographic profile and $^{10}$Be concentration profile can be matched well is shown
(shaded pink, Fig 10a). This zone corresponds directly to the most likely accepted samples (Fig. 9). Examples of model
outputs across this zone all show a good fit to the topographic profile and $^{10}$Be concentration profile (Fig. 10d). The reduced
area covered by the pink shaded region demonstrates how optimising to multiple datasets has constrained uncertainty on
final best-fit parameter results considerably.

Results from topographic-only optimisation (shaded blue, Fig. 10e) show the same declining trend in cliff retreat rates, but
are overall consistently faster than the rates of cliff retreat generated from multi-objective optimisation (shaded pink, Fig.
10e). Furthermore, uncertainty in topographic-only optimised retreat rates is much greater compared to multi-objective
optimisation results, particularly in modern-day modelled cliff retreat rates. In contrast, results from $^{10}$Be concentration-only
optimisation (shaded orange, Fig. 10e) produce slower cliff retreat rates when compared to the multi-objective optimised
results, but still show the declining trend. Multi-objective optimised results (shaded pink, Fig. 10e) show the declining trend
in cliff retreat rates, where the magnitudes are intermediate to the two single-objective function results. Importantly, the
range of retreat rates from multi-objective optimisation is considerably smaller in comparison to single-objective
optimisation. Present-day cliff retreat rates based on multi-objective optimisation range from 1.8-2 cm yr$^{-1}$ in contrast to $^{10}$Be
concentration-only cliff retreat rates of 1-1.5 cm yr$^{-1}$ and topographic-only optimisation cliff retreat rates of 3-9 cm yr$^{-1}$.

Differences between cliff retreat rates on the scale of cm yr$^{-1}$ may not appear noteworthy, but projecting these retreat rates
across large temporal scales have a significant effect on the modelled rock coast erosional history. The 250 m, modern-day
intertidal shore platform would have been eroded in the past 2700-5200 years based on topographic only optimisation, 6800-
7000 years based on $^{10}$Be concentration only optimisation, and 5400-5600 years based on multi-objective optimisation. For
these examples, the time period of modelled cliff erosion that reflects a good match to the measured topographic profile
result in an uncertainty of 2500 years, while an uncertainty based on multi-objective optimisation is only 200 years. The
example rock coast site at Bideford is a stable coastline with relatively slow historic cliff retreat rates. It is important to
ensure modelled cliff retreat rates can be constrained as much as possible, because the magnitude and uncertainty in cliff
retreat rates would increase by orders of magnitude when applying this model to more dynamic rock coast sites, such as the
southern UK coast chalk cliffs that are currently retreating at a much faster rate (eg. Hurst et al. 2016).



Multi-objective optimisation ensures we optimise both objective functions simultaneously, and that best-fit results will
reflect the parameter space where both measured datasets can be matched well. Consequently, equifinality in best-fit results
can be limited substantially, which, most importantly, results in better-constrained modelled cliff retreat rates. Further
improvements to the model optimisation may be made with future inversion calculations by optimising to a third measured
dataset, including a secondary CRN concentration profile (eg., $^{26}$Al or $^{14}$C). This has the potential to 1) further constrain
modelled long-term cliff retreat rates and 2) reveal more complex shore platform erosion evolution through coupled CRN
analysis, e.g., such as platform burial due to sediment cover.

Earth **Surface**
**Dynamics**
Discussions



**Figure 10: Objective function surface at Bideford (a), for parameters material resistance (*b*) and weathering rate (*c*), and material resistance (*b*) and wave height decay rate (*a*). Blue shaded region (a) shows where in the parameter space the modelled topographic profile alone can match measured data. Blue diamond's (a) correspond to example model results shown in (b). Orange shaded region (a) shows where in the parameter space the modelled $^{10}$Be concentration profile alone can match measured data. Orange triangles (a) correspond to example model results shown in (c). Pink shaded region (a) shows where in the parameter space both the modelled topographic profile and $^{10}$Be concentration profile can match measured data. Pink crosses (a) correspond to example model results shown in (d). Cliff retreat rates (m yr$^{-1}$) on a log-scale for each set of model examples are shown in (e).**






**5.2 Parameter correlation**

As previously mentioned, Matsumoto et al. (2018) found that similar modelled topographic profiles can be produced across a wide range of wave force in relation to weathering, particularly in mega-tidal settings. Bideford is situated in a mega-tidal setting with a mean spring tidal range of 8.41 m. As expected, results from the MCMC analysis show accepted samples are

found across a wide range of wave height decay rate (*a*) and weathering rate (*c*) values for Bideford (Fig. 7c; Fig. 8c; Fig. 9c).

The objective function space for wave height decay rate (*a*) and material resistance (*b*) reveals a trade-off relationship between the two functions. A linear regression analysis was performed and highlights a finite range of linearly related *a* and

*b* values can produce a well-matched topographic profile and resultant $^{10}$Be concentration profile. The residuals of *a*, after fitting to the linear correlation in Figure 11a are shown (Fig. 11b). If wave height decay rate (*a*) is increased too much and waves dissipate energy too quickly, then modelled topographic profiles become too steep to match the gradient of the shore platform measured at Bideford (Fig. 11c). If wave height decay rate is decreased too much and waves dissipate energy across a wider distance of the shore platform, the gradient of the modelled topographic profiles become less than 0.02 measured at

the Bideford shore platform (Fig. 11e). The observed range of residuals across the *b/a* regression and the resultant model outputs highlights the narrow uncertainty of *y* required to produce a matching topographic and $^{10}$Be concentration profile.

In order for an MCMC analysis to produce effective posterior distributions, the optimisation method requires free parameters to be independent of each other. As a result of the correlation revealed between *a* and *b* parameters, the high confidence

placed on *a* values (Fig. 11) is not reflected by the posterior distributions produced from the MCMC results (Table 3). Wide posterior distributions of the accepted samples (axis histograms in Figure 11a) produce large uncertainty for final MCMC results. We argue that propagating MCMC uncertainties for *a* together with the uncertainty for *b* produces unrealistic errors in model outputs, specifically seen in the large range of shore platform gradients. Consequently, the uncertainty on final model outputs (Fig. 5; Fig. 6) are constructed by plotting the model result of the median and 16-84% confidence range for

each parameter against the median result for the other two parameters.

Scalby is located at a meso-tidal coastline with mean spring tidal ranges of 4.6 m and does not, however, show distinct correlation between *a* and *b*. Best-fit *a* values are constrained by the lowest bound of *a* for Scalby, where *a* limits are informed by field-based studies (Ogawa et al., 2011). Future investigations into how *b* vs *a* relationships may change as a

function of tidal range within this exploratory model that are informed with additional site-specific datasets are needed in order to understand this model behaviour further.





**Figure 11: (a) Objective function space of accepted samples from the MCMC analysis for material resistance (*b*) and wave height decay rate (*a*) parameters for Bideford. A linear regression calculation was performed and shown along with the r value of -0.83. Histograms on either axis show the distribution of accepted sample points. (b) Residual plot from the regression line shown in (a). Yellow circles plotted along the regression line in (b) correspond with model outputs shown in (d) that show the best-fit to the measured topographic profile. Green triangles pointing upwards (b) are plot +0.2 from the regression line and correspond with model outputs shown in (c). Green triangles pointing downwards (b) are plot -0.2 from the regression line and correspond with model outputs shown in (e).**




## 5.3 Equifinality in cliff retreat trajectories

Correlation between *a* and *b* parameters at Bideford also shows that a matching topographic profile can be produced across the full extent of material resistance (*b*), provided wave height decay rate *(a)* has adjusted accordingly (Fig. 11d). A greater
material resistance (*b*) requires a greater wave force i.e., smaller wave height decay rate (*a*) in order to erode an across-shore intertidal shore platform with the same geometric properties. Correlation between *a* and *b* demonstrates one potential source of equifinality in terms of endmember topographic and [10]Be concentration profiles. To ensure we report accurate cliff retreat rates, we need to identify whether equifinal *a* and *b* combinations result in similar cliff retreat trajectories as well as the same endmember model topographic and [10]Be concentration profiles. The magnitude of material resistance (*b*) across the *b/a*
regression has no effect on the final fit for the topographic profile (Fig. 11d). Most importantly, the resultant cliff retreat rates all show comparable trajectories and rates of cliff retreat across the late-Holocene. Therefore, as long as the combination of *a* and *b* track the regression fitted to the accepted samples (Fig. 11a), we can have confidence that the most accurate retreat rates are reported.

## 5.4 Interpretation of best-fit parameter values

It is important to recognise that the best-fit lower wave height decay rates (*a*) found at Scalby, which results in a breaking wave force to be exerted over a greater distance across the platform surface, do not necessarily mean that wave energy is greater at Scalby than Bideford. Wave energy is abstractly defined in the coastal evolution model by an assailing wave force, and we have chosen to explore this in the MCMC analysis by varying the wave height decay rate (*a*). Our aim is not to
quantify the wave force at real-world coastlines, but to determine the best combination of model parameters to match measured datasets in order to model cliff retreat over timescales much longer than information on wave conditions is available. Wave height attenuation length is dependent on other factors such as profile gradient and tidal range, which are site-dependant (Trenhaile, 2000). We also have very few constraints on if and how tidal range has changed across the past 8,000 years, as this depends strongly on local and far-field bathymetry, and other uncertain climatic variables. Furthermore,
wave height decay rate is further impacted by surface roughness (Poate et al., 2018), and this is not considered within our model simulations. For these reasons, using wave height decay rate to infer wave energy for a range of real-world sites is unachievable with our model, even with a rigorous optimisation method. Moreover, parameter correlation, which seen particularly well in the relationship between material resistance (*b*) and wave height decay rate (*a*) (see section 5.2), means that finding a single best-fit result for wave height decay rate (*a*) is problematic without isolated parameter investigations.
Nevertheless, the relative importance of the parameters that control the wave driven erosion (*a*) and weathering controlled erosion (*c*) can be considered. Because best-fit results at Bideford clearly show that negligible contributions of weathering are needed to produce the [10]Be concentration profile, we can conclude that the evolution of cliff retreat at Bideford is likely



dominated by wave-driven erosion. In contrast, modelled best-fit values of low to intermediate weathering rates at Scalby
reveal a more complex interplay between wave-driven and weathering-controlled erosion.


Resultant equifinality in cliff retreat trajectories (Fig. 11) reveal that correlation does not prevent identification of consistent
patterns in cliff retreat rate histories. In other words, the relative combinations of *a*, *b* and *c* parameter values are able to
capture the morphodynamics needed to model compatible rock coast evolutions at unique rock coast sites. Therefore, the
abstract representation of wave force, weathering processes, and material resistance within the RPM does not inhibit the
modelling of cliff retreat for real-world sites.

**6 Conclusions**

In this study, we have developed a multi-objective optimisation approach to reconstruct the history of rock coast evolution
through the combination of morphodynamic modelling and field observations. Our approach calibrates a coupled
morphodynamic-cosmogenic radionuclide rock coast evolution model using observations of modern rock coast topography
and measurements of *in situ* $^{10}$Be concentrations in the exposed bedrock. These developments are vital to enable application
of a process-based model to real-world coast sites and quantify a time series of rock coast erosion and sea cliff retreat rates.
Our results demonstrate the necessity for using multi-objective optimisation in order to limit model equifinality, in which
similar topographies develop via differing evolutionary trajectories. Optimal parameter selection is used to minimise
discrepancies between model simulations and measured topography and $^{10}$Be concentrations can reveal the most likely
history of rock coast development, including rates of shore platform lowering and cliff retreat.

The selection of free parameters within the model optimisation focuses on the efficacy of intertidal weathering and erosion
processes relative to the resistance of bedrock. There is still equifinality in model outcomes for parameter combinations
where similar patterns of topographic development occur. More resistant bedrock combined with efficacious
weathering/erosion can result in development of a rock coast profile similar to that of a less resistant bedrock and less
effective weathering/erosion. This parameter correlation can be reduced through multi-objective optimisation, but ultimately
does not prevent identifying consistent patterns in cliff retreat rates at specific rock coast sites.

Investigations into the Pareto set of optimised results show best-fit results are consistent across a range of objective function
weightings. These findings suggest that a single, equally weighted MCMC chain is sufficient to find an optimal set of input
model parameters in order to constrain the cliff retreat rate history.

By applying rigorous optimisation methods, we demonstrate how an abstract, process-based model is capable of replicating
rock coast profile development of real-world sites. Moreover, when coupled with a $^{10}$Be production model, equifinality is
constrained sufficiently to reveal distinctive trends in long-term cliff retreat rates. Long-term cliff retreat rates of two unique

UK rock coast sites both closely mirror the history of RSL change rates. These findings indicate that future accelerations in RSL rise associated with climate change will cause accelerations in cliff retreat rates, even at coastal sites that have been stable historically. We are only able to understand how cliff retreat responds to RSL by modelling the trajectory of cliff evolution across timescales that capture these changes in RSL rise.

The multi-objective statistical modelling approach developed and tested in our study highlights potential for future efforts to 1) reconstruct past rates and patterns of cliff retreat over timescales appropriate to the magnitude and frequency change of erosion at rock coast sites, 2) assess the relative importance of weathering and wave-driven erosion processes, and 3) forecast future erosion rates under different scenarios for RSL change as a result of climate change.

**Code and data availability.** Input datasets are presented in the paper and/or the supporting information and will be made available from public sources specified. Output data and plotting scripts used to create the figures for this paper are currently available from Shadrick et al., (2021). All data and code used in this analysis will be open source and will be downloadable from a github repository (in progress): https://....githib.io// but can be requested from Shadrick et al., (2021) presently.

**Author contributions.** J.R.S., M.D.H., M.D.P. and D.H.R. designed the research and analysed the data; J.R.S., D.H.R., M.D.H., B.G.H., K.M.W. and A.J.S. collected measured datasets; D.H.R., J.R.S., B.G.H., A.J.S., prepared samples in the laboratory; J.R.S. developed the optimisation routine with support from M.D.H and M.D.P.; K.M.W. performed AMS measurements; J.R.S., M.D.H., D.H.R. and M.D.P. wrote the paper.

**Competing interests.** The authors declare that they have no conflict of interest.

**Acknowledgements.** This work was supported by a studentship from the Natural Environment Research Council (NERC) Science and Solutions for a Changing Planet Doctoral Training Partnership (DTP), with additional funding by the British Geological Survey (BGS) to J.R.S. We acknowledge the support of ANSTO Research Portal award 10955 to D.H.R. We thank Michael Ellis for general and ongoing support with the project and editorial assistance. We would also like to thank Katy Barnhart for invaluable assistance with the Dakota software implementation and Geraldo Fiorini Neto for help with Dakota installation and testing. We also thank Sarah Bradley for providing RSL history data, which is an essential component to this project. Finally, we would like to thank Emily Gusterson and Jay Ward for support with field data collection.



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
