# Peer review of "Multi-objective optimisation of a rock coast evolution model with cosmogenic 10Be analysis for the quantification of long-term cliff retreat rates"

_Earth Surface Dynamics, 2021_

## Referee Comment (RC2)

**1  Summary**

I appreciate the opportunity to review Shadrick and others (submitted to Earth Surface Dynamics, 2021). In this study the authors estimate millennial scale cliff erosion rates using a coupled model of rocky coast evolution and cosmogenic radionuclide production. The authors implement a multi-objective optimization approach to understand the relative roles of $^{10}$Be concentrations and topographic profiles on constraining model parameters. Because the authors consider two sites, they are able to provide insight into the extent to which results are general or site specific. The study is well designed and well described. The work reflects a novel application of optimization to interpreting geologic data and it represents a big advance in using this type of methodology to interpret geologic data and geomorphic models. The authors nicely explore a variety of important topics including the relative contribution of different types of data to parameterize a coupled model, and the role of parameter covariance and equifinality.

I recommend it for publication after minor revision.

**2  Narrative comments**

1. Provide more explanation and/or justification for the 8 kyr model duration. Was this chosen because it is the extent of GIA modeling, needed for RSL as a boundary condition, or because of some other reason? On this same point, I'd recommend providing a bit of context to the reader regarding typical timescales to reach a steady state topographic profile under steady forcing. It is clear that this is not your aim, since you use nonsteady RSL forcing, and will of course depend on parameter values... however, it will be helpful context for how far away from equilibrium the coastal profile would be over a 8 kyr duration.

2. Similarly, provide additional information regarding what initial conditions were used in the model, and whether the model shows sensitivity to the initial conditions over the timescale simulated.

3. Recommend that most of section 3.4 (MCMC inputs) be moved into section 3.2 (The coastal evolution model). Specifically, I would recommend adding to section 3.2 an introduction to the parameters $y$ and $K$ (as $F_R$ is already introduced) as well as a brief description of other parameters in the model presented by Matsumoto, Dickson, and Kench (2016) which are not considered here. It is certainly reasonable to not consider these parameters, but let the reader know a bit more about what they are.

   I would then recommend putting most of section 3.4 at the end of a revised section 3.2. This will help the reader understand what the model and parameters are before you begin section 3.3 and discussion of Dakota.

4. I was surprised not to see a plot of the pareto front itself (that is, a plot with the scaled and weighted topographic RMSE on one axis and the $^{10}$Be RMSE on the other axis, with one line for each of the two sites). The simulated topography and $^{10}$Be concentrations are provided in Figure 4, but the pareto trade-off itself is important to visualize in a multi-objective function study. As the multi-objective nature of this study is so novel, I'd strongly recommend adding such a subplot to Figure 4. It would support the text from lines 435–448, namely that the hump in the Bideford $^{10}$Be data is best fit only when the topographic fit is mostly ignored.

5. None of the simulations seem to capture well the increase in slope in the most shore-ward 50 m of the the Scalby site. Could you discuss this more? Are there field observations from Scalby that might provide more context to this topographic feature?

6. Your results show that cliff retreat rates match RSL rise closely. But they also show (Figure 6) that scaled RSL is on the lower end of the Bideford retreat rates, while it is on the upper end of the Scalby rates (dashed line is below(above) the mean line for Bideford (Scalby)). Is this an interpretable result?

Are there differences in the wave climate, geomorphology, or geology of the sites that could explain this?

7. You nicely discuss parameter covariance near the end of the manuscript. However, there were two points related to parameter covariance that I think are important to discuss. First is related to your specific implementation. You set $K = 5^c F_R = 5^c 10^b$, removing some parameter covariance that would have existed otherwise. Why not treat them as fully independent and document the nature of the covariance as you do for $a$ and $b$ in section 5.2.

   A second, bigger picture comment that you are well poised to make is about the "meaning" of these model parameters. Namely, when model authors write models, we often think of the parameters as independent and meaningful—and sometimes linked to field or laboratory measurements (Dietrich et al. 2003). But it is not uncommon to find these sort of covariance issues, most often because of how these parameters are used. In the case of the model presented by Matsumoto, Dickson, and Kench (2016) is this something that could be anticipated because of the mathematical form of the model? Or is it fully emergent.

8. One theme that emerges from both sites is the overwhelming influence of the RSL boundary condition. Looking at the different cliff retreat rates in Figure 4 it seems reasonable to conclude that no matter the topography vs $^{10}$Be weighting the estimated cliff retreat rates match the RSL forcing. It might be worth commenting in the discussion on the relative role of the boundary condition vs parameter values for this type of analysis.

9. Overall all figures are well designed and clear. However, my black and white printer led to this comment: consider a grayscale safe color scales (e.g., viridis used in Fig 9) and/or different symbols for Bideford and Scalby sites. Colorbrewer is a good resource for this.

**3    Line level comments**

Bullet points in this Section indicate "<LineNumber>", "T<Table Number", or "F<Figure Number>".

15 Perhaps add a statement about the sort of timeframes that would be possible without this method. Such a statement would make clear by contrast the benefit of this approach.

185 The model represents a cross-section, yes? Recommend using the term "cross section" if this is the case.

220 At the end of this paragraph I wanted a sentence or two introducing the concept of the pareto front.

231 I think you need to add something like the phrase "with different weights" to the end of this sentence, because it is specifically the nature of the different weights that allows you to explore the pareto front.

235 I find it helpful to include a section or subsection "Model Implementation" that puts together information like the cell size, as well as a few items not stated. For example, what timestep was used? This would also be where you might state the simulation duration and why it was chosen.

242 You mention measurement error here, but it is not clear whether measurement error was incorporated into the RMSE. In your case, I'd expect you could have a different measurement error for each $^{10}$Be observation, and that including/excluding this might have an impact on the Bideford results because the highest $^{10}$Be measurements also have the highest error.

246 Why is $w_i$ in a square root? I would expect just $w_i$ here such that $\sum_{i=1}^{N_i} w_i = 100$.

255 Your discussion and approach to scaling makes sense when I read it here. However, after reading this portion of the methods I was confused when I got to Figures 7, 8, and 9 because I was expecting the topographic and $^{10}$Be values to be of similar magnitude (which they are not). I think my confusion could be addressed with minimal revision to sections 3.3.2 and 3.3.3. Specifically, I would recommend talking only about constructing the topo and $^{10}$Be RMSE values in the first of these sections and introduce an equation that looks something like

$$RMSE_i = \sqrt{\frac{1}{N_j} \sum_{j=1}^{N_j} \left( \frac{Mod_{i,j} - Meas_{i,j}}{\sigma_{i,j}} \right)^2} \tag{1}$$

Here I've also added a term for measurement error $\sigma_{i,j}$. Then, after discussing how the two RMSEs were each constructed, in the second section introduce scaling/weighting and constructing the pareto front with an equation like

$$TotalRMSE_p = \sum_{i=0}^{N_i} \frac{w_{i,p}}{s_i} RMSE_i \tag{2}$$

This would more clearly separate the construction of your two objective functions from the scaling and weighting of them for the pareto analysis since you don't need to discuss scaling/weighting until you get to the discussion of multiobjective and the pareto front. Similarly, I would recommend adding a subscript of some sort (I've used $_p$ above) to denote that the weights change depending on which pareto set is being used.

317 Recommend introducing $y$ and $K$ earlier, as discussed in the narrative comments.

392 This is the first point in the text where the simulation duration of 8 kyr is mentioned. I would recommend mentioning it earlier as well as providing context regarding why that duration was used at that point in the text.

458 Since you only consider one point on the pareto front, I think this section title is misleading. Consider revising.

519 The last sentence of the figure caption is confusing to me. I think you mean to say that the gray shaded region corresponds to cliff retreat rates which occurred when the cliff was further offshore than your topographic profile measurements. Revise for clarity.

683–685 Recommend framing this differently. Rather than making a statement about confidence, I'd recommend pointing out that it is rare to formally evaluate how different data sources ($^{10}$Be, topo) constrain a model differently. Each of these are valid data, and if you had only one (most commonly, no $^{10}$Be) it would be totally fine to parameterize the model with only the available data (see, for example, any paper fitting river long profiles to some sort of fault history and/or value of fluvial erosion coefficient). But because you have both types of data you have the opportunity to evaluate the relative *information* provided by each data source. If, as is not the case here, both data sources yielded the same parameter estimates, we would learn we don't gain anything from the second dataset (and it is thus not necessary to make those observations). In contrast, as you find, if the two data sources contain different information, these datasets pull the coupled model in different directions. An potentially relevant reference here is Furbish (2003).

T1 Are the weights listed $w_i$ or $\sqrt{w_i}$. Please clarify.

T2 Why is the base for $K$ 5 rather than 10?

F3 I'd recommend adding one more loop here to denote that this workflow is done for each location on the pareto front. You nicely emphasize how your approach could be generalized to include additional objective functions, and this could help emphasize that it can be generalized to additional points on the pareto front.

F5 Because the right column represents a zoom into a portion of the left column, a gray rectangle or similar that indicates this region in the left column would guide the reader.

F6 Similarly a box in the upper right of each panel showing the extent of the inset would be helpful.

F6 Why only show 7 kyr when the simulation duration is 8 kyr. If the first 1 kyr is a "burn in period" to forget the initial conditions, that is reasonable but should be stated.

F7,8,9 Rather than plotting the objective function itself in Figures 8 and 9 I would recommend plotting the posterior. It is often easier to interpret because it does not have the same issues with overplotting that the objective function provides. One nice tool for this is corner.py (Foreman-Mackey 2016). It nicely also shows the marginal distributions.

F10 It is not clear if these three different sets (orange, blue, pink) come from different pareto sets, or different samples from the 50-50 evaluation. Clarify.

756 This is a very nice subsection and analysis.

**References**

Dietrich, W. E., D. G. Bellugi, L. S. Sklar, J. D. Stock, A. M. Heimsath, and J. J. Roering (2003). "Geomorphic Transport Laws for Predicting Landscape form and Dynamics". In: *Prediction in Geomorphology*. American Geophysical Union (AGU), pp. 103–132. DOI: https://doi.org/10.1029/135GM09. eprint: https://agupubs.onlinelibrary.wiley.com/doi/pdf/10.1029/135GM09. URL: https://agupubs.onlinelibrary.wiley.com/doi/abs/10.1029/135GM09.

Foreman-Mackey, D. (2016). "corner.py: Scatterplot matrices in Python". In: *Journal of Open Source Software* 1.2, p. 24. DOI: 10.21105/joss.00024. URL: https://doi.org/10.21105/joss.00024.

Furbish, D. J. (2003). "Using the Dynamically Coupled Behavior of Land-Surface Geometry and Soil Thickness in Developing and Testing Hillslope Evolution Models". In: *Prediction in Geomorphology*. American Geophysical Union (AGU), pp. 169–181. DOI: https://doi.org/10.1029/135GM12. eprint: https://agupubs.onlinelibrary.wiley.com/doi/pdf/10.1029/135GM12. URL: https://agupubs.onlinelibrary.wiley.com/doi/abs/10.1029/135GM12.

Matsumoto, H., M. E. Dickson, and P. S. Kench (2016). "An exploratory numerical model of rocky shore profile evolution". In: *Geomorphology* 268, pp. 98–109. DOI: https://doi.org/10.1016/j.geomorph.2016.05.017. URL: https://www.sciencedirect.com/science/article/pii/S0169555X16303385.

---

## Author Response (AR1)

Replies are in *BLUE* with text quoted from the revised manuscript in *GREEN.*

Line numbers refer to the revised manuscript.

*We would like to thank both reviewers for their supportive and constructive*
*feedback. Both have recognised the value in the methods developed, which is highly*
*encouraging. We feel that the revisions made based on both reviewers' comments*
*strengthen the manuscript greatly.*

Reviewer #1 (Vincent Regard):

The paper entitled "Multi-objective optimisation of a rock coast evolution model with
cosmogenic 10Be analysis for the quantification of long-term cliff retreat rates" by Jennifer
Shadrick and colleagues, reports on work aimed at understanding how the joint recording of
rock platform topography and the cosmogenic isotopes (10Be in this case) of the rocks that
constitute it can provide a good description of the history of coastal cliff retreat during the
Holocene. The work is based on a model of the evolution of the rock platform, associated
with a module describing the enrichment in cosmogenic isotopes. This direct model requires
knowledge of a number of variables, in particular the erodibility of the rocks (FR), the wave
dissipation coefficient on the platform (y) and the weathering of the rocks in the intertidal
zone (K). This model is run a number of times via an optimisation procedure based on the
RMSE of the difference between model and data. This optimisation is proposed for two
different English sites: Scalby (NE) and Bideford (SW), for which a rate of recession over time
is produced, which shows a very clear correlation with the sea level rise rate. The discussion
then turns to the combined effects of the 3 variable parameters.

I strongly support the publication of this work, although I have some formal reservations,
which I express below.

*We would like to thank Vincent Regard for his highly constructive and thoughtful*
*review. We are happy to see that his review is overwhelmingly positive and that he*
*shares our enthusiasm about the significance of our work.*

**Strong points**

Innovative inversion work, which advances knowledge of the problem. The tools developed
will be available to the community, I hope.

Good data

The result in Figure 7 is excellent, although there is probably room for discussion.
*We have addressed this comment on line 355 of this response document.*

In the evolution of rock platforms, two erosive drivers are compared here that have never been compared before: erosion by waves vs. weathering in the intertidal domain. Indeed, this weathering has only been documented on the basis of laboratory experiments (Kanyaya and Trenhaile, 2005, Porter et al. 2010). This work is to be commended for advancing the comparison between wave erosion and weathering.

**Weaknesses**

The description of the numerical process is difficult, not always understandable by an outsider (especially a non-numerician such as myself). There is also a mix of details (e.g. line 185-186) and very general considerations (e.g. description of the model of Matsumoto et al. 2016 without resolution, time step). I think the presentation of the methodology needs some work. The parameters a, b, c are not understandable by the text alone (Table 2 is needed). Some very long sentences are a bit complex to understand for a non-native English speaker.

*We appreciate that model description is difficult to follow in certain sections. However, we have purposely simplified and included a shortened description as the stand-alone model is fully explained in previous literature, e.g., (Matsumoto et al., 2016, 2018) and Hurst et al., 2021 (In prep).*

*Nevertheless, we agree that more can be done to clarify model description and numerical processes. Below are all examples of where we've attempted to simplify and clarify the methodology:*

1) *An additional schematic figure has been added to help clarify the model description (previously line 185-186), and show gridded framework, for example.*

[Figure]

*Line 339-345: (Figure caption) "Figure 3: Coastal evolution model schematic.*
*Topographic profile cross-section constructed in a gridded framework, showing*
*wave approach and tidal duration distribution. Binary values of 0 and 1 are*
*assigned to water/air and rock categorised cells respectively. Rock cells (value 1)*
*are eroded and removed from the profile (assigned value 0) once wave force*
*exceeds material resistance ($F_W \geq F_R$) (b,c). Subaerial weathering (K), can also*
*lower the material resistance value ($F_R$). Wave height decay rate (y) controls the*
*distance waves can break across the shore platform and as a result, the erosional*
*potential of wave assailing force $F_W$."*

*2) The modelled time step was previously mentioned, but we have added a clearer*
*statement and explanation. This has been included in the newly added section:*
*3.2.1. Model implementation, as suggested by reviewer #2 (see below). Added*
*details of cell resolution etc. which should be aided with new figure (see above).*

*Line 351-374: "3.2.1 Model implementation*
*Other fixed model parameters and initial model conditions are set to the same*
*values as used by Matsumoto et al. (2018) (supplementary materials Table S7).*
*Once the model burn-in period has been exceeded (first ~1000 years), the initial*
*conditions, such as platform gradient, have negligible effect on final outputs of*
*topography, $^{10}$Be concentrations and retreat rates. The RSL history input is taken*
*from the GIA model of Bradley et al. (2011). RSL uncertainty was not considered as*
*we expect it to make little difference to final results. For southern UK sites across*
*the late-Holocene, the misfits between measured RSL data and GIA model*

*predictions are minor (Bradley et al., 2011). Uncertainties of magnitude $\pm 0.01$-$0.1$*
*mm y$^{-1}$ of RSL rise have negligible impact due to the spatial and temporal*
*resolution considered for the model. A fixed mean spring tidal range of 8.41 m for*
*Bideford and 4.6 m for Scalby are used, which are based on tide gauge records*
*(National Tidal and Sea Level Facility, 2021).*

*We chose to implement a model simulation time of 8000 years. A simulation time of*
*8000 years BP to present day captures the RSL history curve for both sites (Fig. 2),*
*where rapid RSL rise occurs for the first ~1000 years, followed by a slow decline*
*from 7000 years BP to present day. So, we can observe how cliff retreat rates will*
*respond to these different stages in the RSL history. Implementing a simulation*
*time of 10000 years, for example, would have no impact in the late stages of the*
*simulation, such that there are no differences in optimisation.  A simulation of*
*10000 years would also increase the computer run time unnecessarily.*
*Furthermore, previous studies show that under static RSL conditions, a steady-state*
*equilibrium is reached, where cliff retreat rates stabilise after rapid initial retreat,*
*by 8000 years (e.g. Walkden and Hall (2005)). Modelling rock coast evolution*
*across an 8000-year window means only a Holocene history for shore platform*
*formation has been considered, with no possible re-occupation from a previous*
*interglacial period (e.g. Choi et al. (2012)). The $^{10}$Be concentration datasets used to*
*develop this optimisation routine at both sites exhibit low concentrations,*
*suggesting these rock coast features are Holocene-formed (Regard et al., 2012).*
*Therefore, these datasets are suitable for modelling Holocene-formed shore*
*platforms, as a means to develop this optimisation routine. During the 8000-year*
*simulation time, the topographic profile and $^{10}$Be concentrations are calculated and*
*output every year (1-year timestep). The model space is split into 10x10cm gridded*
*cells (Fig. 3)."*

*3) In the text, description of parameters a, b and c have been included, as well as*
   *in table 2. The section 3.4 MCMC analysis inputs, now only includes*
   *information on the choice of free parameter ranges, as parameter description*
   *has already been included in section 3.2.*

*Line 626-628: "$y = 10^a$*
*$F_R = 10^b$*
*$K = 5^c \times F_R$"*

*4) Where we felt appropriate, technical language has been simplified, e.g.*

*Line 189-190: "The exploratory model uses a grid framework, in which cells are*
*assigned a binary value of 1 (rock) or 0 (water/air), and represents a cross section*
*transect (elevation and distance), taken perpendicular to the cliff line (Fig. 3)."*

Two other parameters would have deserved to be considered as variable (i.e. not perfectly
known): the sea level history, or the incident waves. I think it is a bit late to integrate them
into this work, but it would be interesting to mention them, if only qualitatively.

*We have added a statement as to why we have not chosen to vary incident wave*
*height and sea level history, and what influence we would expect from varying*
*these model inputs. Relation between wave height and material resistance is*
*already understood in the model – more resistant rock needs higher wave height to*
*achieve same amount of erosion. Focus on wave height decay rate, will explore the*
*process dynamics and resultant morphological outcome.*

*Line 355-358: "RSL uncertainty was not considered as we expect it to make little*
*difference to final results. For southern UK sites across the late-Holocene, the*
*misfits between measured RSL data and GIA model predictions are minor (Bradley*
*et al., 2011). Uncertainties of magnitude $\pm0.01$-$0.1$ mm $y^{-1}$ of RSL rise will have*
*negligible impact due to the spatial and temporal resolution considered for the*
*model."*
*Line 576-582: "Wave erodibility is explored in the MCMC analysis by varying the*
*wave height decay rate (y), which is consistent with previous modelling approaches*
*(e.g., Matsumoto et al., 2018; Trenhaile, 2000). Incident wave height is kept*
*constant throughout model simulations. We chose to explore wave erodibility in the*
*model by varying wave height decay rate (y) over incident wave height, as a linear*
*relationship between input wave height and material resistance ($F_R$) is already*
*established: greater wave height needs to be compensated by an increase in*
*material resistance ($F_R$) (Matsumoto et al., 2016). Whereas, by focussing on process*
*dynamics with wave height decay rate (y), the spatial distribution and degree of*
*wave erosion can be considered; this will have implications for the evolving shore*
*platform morphology."*
I still have a second order question: how to explain the group of points >180m from the cliff
at Bideford: is there an expression on the platform explaining these points that stand out
from the others?

*We agree this is an interesting observation, which may lead to an important*
*discussion on how the model represents erosional processes across the shore*
*platform. We, however, feel this discussion point is beyond the scope of this study as*
*we want to focus on the methodologies developed and not interpretation of the*
*measured data, that here, acts only as test datasets. A secondary paper is in*
*development that focusses on the site-specific geomorphological interpretations of*
*the model best fit results.*

**Conclusion**

I am very supportive of the publication of this work. I hope that my comments will help to
improve it through moderate modifications, as well as open up perspectives for further
investigations.
**Other remarks**
Line 36-37. Premaillon et al. Esurf 2018 could be cited here.
*Citation added*
*Line 37-40: "Thus, the processes that effect the weathering, erosion and transport*
*of shore platform, intact cliff, failed cliff and other beach material are an important*
*part of the whole process of 'cliff erosion' (Coombes, 2014; Hurst et al., 2016;*
*Limber and Murray, 2011; Masteller et al., 2020; Naylor and Stephenson, 2010;*
*Prémaillon et al., 2018; Thompson et al., 2019)."*
Field location. The authors should present the sites a little better. I suggest a photo of each
site, especially so that the reader understands the influence of the geological structure on
the morphology of the platform/cliff system
*As previously mentioned above, the focus of this study is not the measured datasets,*
*or specificities of the field sites, but the optimisation routine. We only wanted to*
*include a site location map for context but think site photos are unnecessary for the*
*purpose of this paper. The geomorphological interpretation of the optimisation*
*results will be included in the previously mentioned secondary publication, this will*
*include a more detailed study site figure with site photos that show the geological*
*structure.*
Figures: the uncertainties shown by the shaded areas are unreadable. The colours should,
for example, be reinforced.
*Shaded areas of figures darkened as suggested to make uncertainties clearer.*
Line 198. More details on the model would be welcome: time step, spatial resolution. Which
tide range did you use: the spring one, an average one?
*Added more model details as requested: time step 1 year, 8000-year simulation time,*
*spatial resolution. We did already mention we used the mean spring tidal range,*
*this likely got lost in the MCMC input section, but now this information is placed in*
*the Model implementation section, so should be made clearer to the reader.*
*Line 351-374: "3.2.1 Model implementation*
*Other fixed model parameters and initial model conditions are set to the same*
*values as used by Matsumoto et al. (2018) (supplementary materials Table S7).*
*Once the model burn-in period has been completed (first ~1000 years), the initial*
*conditions, such as platform gradient, have negligible effect on final outputs of*
*topography, $^{10}$Be concentrations and retreat rates. The RSL history input is taken*

*from the GIA model of Bradley et al. (2011). RSL uncertainty was not considered as*
*we expect it to make little difference to final results. For southern UK sites across*
*the late-Holocene, the misfits between measured RSL data and GIA model*
*predictions are minor (Bradley et al., 2011). Uncertainties of magnitude $\pm 0.01$-$0.1$*
*mm $y^{-1}$ of RSL rise have negligible impact due to the spatial and temporal*
*resolution considered for the model. A fixed mean spring tidal range of 8.41 m for*
*Bideford and 4.6 m for Scalby are used, which are based on tide gauge records*
*(National Tidal and Sea Level Facility, 2021).*

*We chose to implement a model simulation time of 8000 years. A simulation time of*
*8000 years BP to present day captures the RSL history curve for both sites (Fig. 2),*
*where rapid RSL rise occurs for the first ~1000 years, followed by a slow decline*
*from 7000 years BP to present day. So, we can observe how cliff retreat rates will*
*respond to these different stages in the RSL history. Having tested longer*
*simulation times, implementing a simulation time of 10000 years, for example,*
*would show no change to final model outputs for nearshore topography or $^{10}Be$*
*concentrations. Our results are thus independent of this initial boundary condition,*
*and longer simulations would increase the computer run time unnecessarily.*
*Modelling rock coast evolution across an 8000-year window means only a*
*Holocene history for shore platform formation has been considered, with no*
*possible re-occupation from a previous interglacial period (e.g. Choi et al. (2012)).*
*The $^{10}Be$ concentration datasets used to develop this optimisation routine at both*
*sites exhibit low concentrations, suggesting these rock coast features are Holocene-*
*formed (Regard et al., 2012). Therefore, these datasets are suitable for modelling*
*Holocene-formed shore platforms, as a means to develop this optimisation routine.*
*During the 8000-year simulation time, the topographic profile and $^{10}Be$*
*concentrations are calculated and output every year (1-year timestep). The model*
*space is split into 10x10cm gridded cells (Fig. 3)."*

Part 3.3 is very technical, sometimes hard to understand.

*We apologise that we did not make this section easy to follow. To resolve this, we*
*have simplified the language where appropriate, and have also broken down the*
*single equation into 2 simpler components, as suggested by reviewer #2. We now*
*feel with both reviewers' suggestions, section 3.3. is much easier to follow and*
*understand. See below comments for changes made.*

Line 357. Presentation of a, b, c rather obscure.

*Description of a, b and c changed to an equation style, on separate lines, to clarify,*
*as well as included in table 2.*

*Line 626-628: "$y = 10^a$*
*$F_R = 10^b$*
*$K = 5^c \times F_R$"*

Paragraph 362-375. Here the authors only consider the Holocene history. Is a reoccupation
of an older platform possible? Is it possible to test this hypothesis?

*We have preliminary addressed this in section 3.2.1, model implementation. Here*
*we now justify the simulation time chosen and why we have only considered the*
*Holocene history. We indicate that the low concentrations are suggestive of a*
*Holocene-formed feature. But again, we want to leave site-specific interpretations*
*to the secondary paper, so have only mentioned this briefly here. In the secondary*
*paper we explore the site-specific $^{10}$Be concentration distributions and address what*
*they may indicate for platform erosion, while referencing appropriate literature.*

*Line 367-372: "Modelling rock coast evolution across an 8000-year window means*
*only a Holocene history for shore platform formation has been considered, with no*
*possible re-occupation from a previous interglacial period (e.g. Choi et al. (2012)).*
*The $^{10}$Be concentration datasets used to develop this optimisation routine at both*
*sites exhibit low concentrations, suggesting these rock coast features are Holocene-*
*formed (Regard et al., 2012). Therefore, these datasets are suitable for modelling*
*Holocene-formed shore platforms, as a means to develop this optimisation routine."*

Line 386. I would add that this value of 20mm/yr is unrealistic.

*Added comment as suggested:*

*Line 597-600: "The greatest rate of weathering that we apply when exploring the*
*parameter space for optimisation is equal to: $F_R \times 0.2$ kg $m^2$ $yr^{-1}$, which, results in*
*maximum down-wearing rates of 20 mm $yr^{-1}$ when only considering weathering*
*contributions to shore platform downwear. Rates of 20mm $yr^{-1}$ is unrealistically*
*high for a sandstone platform (e.g. Yuan et al. (2020))."*

Figure 4: There is an error in the unit of the Cliff retreat rates.

*We thank the reviewer for spotting this error. Unit error has been changed.*

Change the grey (50%-50%) to another colour.

*Changed grey colour of 50%-50% results to a contrasting colour.*

Recall why there is cyclicity in the modelled 10Be concentrations.

*Added comment to results highlighting saw-tooth pattern in 10Be is caused by*
*gridded cell resolution, as one cell is removed from the rock array, concentrations*
*will drop as the most abundant, surficial layer of rock is removed.*

*Line 746-748: "The saw-tooth pattern seen in the $^{10}$Be concentration profile is*
*caused by the cell framework resolution of the model. When a surficial rock cell,*
*with greatest $^{10}$Be concentrations, is eroded and removed from the rock profile, $^{10}$Be*
*concentrations drop, as a subsurface cell with less $^{10}$Be is unveiled."*

At Bideford, the cosmo/topo data disagree from 180m away from the cliff. Do you have any
idea why?

*We believe this disagreement in model and measured $^{10}$Be concentration profiles could be explained by the model's absence of spatial variability in platform downwear. In this case, the model looks to be underestimating subaerial platform erosion. This will be a suitable discussion point for the secondary paper previously mentioned. However, we want to avoid site specific interpretations for this study. We have added a sentence to address this.*

*Line 801-802: "Deviations between modelled and measured topography and $^{10}$Be concentrations should be interpreted carefully in the context of local variation in process rates."*

Figure 5. The topo profiles need explanation: this is the current profile, the age corresponds to when the cliff foot was there, but not at the same elevation since there is a downwearing effect.

*We agree with the reviewer's remark here and we have made sure to explain this more clearly by adding an appropriate statement in the text. We have indicated that the timestamps only correspond to the horizontal position of the cliff foot throughout the Holocene, and that the elevation of the cliff-platform junction is not the same due to down wearing.*

*Line 792: "The topographic profiles shown are the present-day positions (Time 0 k yr BP)."*

*Line 806-809: "Time stamps for modelled cliff positions are back calculated and shown for the corresponding distance across the shore platform. For example, the modelled cliff position at Bideford was 200 m offshore from the present-day cliff position ~5000 yrs BP (Fig. 6b). These timestamps correspond to when the horizontal position of the cliff foot was there, but not the exact elevation as down-wearing has occurred since."*

Paragraph 495. Your conclusions in lines 499-501 are tremendous.

*We are delighted that the reviewer is as enthused by these novel results as we are.*

Paragraph 4.3.1. Very important paragraph in my opinion. The results are that waves > weathering. The difference between the two processes is that weathering can sustain cliff recession for a longer period of time while the waves, which dissipate over the rock platform, eventually fade away. This has important consequences: anthropogenic sea level rise will necessarily be accompanied by an acceleration of cliff recession rates.

*We agree with the reviewer here, that this is indeed an important result. This result is the basis for the secondary paper that explores how sea level rise effects cliff retreat rates and focusses on these wave-dominated systems. We have added a statement here to highlight the importance of this result, but again want to leave the full exploration and interpretation to the following paper.*

*Line 962-963: "For both sites, these results strongly imply that wave driven erosion dominates over subaerial weathering in the long-term."*

Figures 10 and 11 are good, but why not provide the equivalent for Scalby?

*We decided not to include the equivalent figures for Scalby because we thought them to be unnecessarily repetitive. We believe the figures for Bideford alone successfully showcase the purpose of the discussion points made in discussion sections. Furthermore, the results for Bideford highlight these points in the most simplified way.*

*For example, please see below an example of part of the figure 10 (now figure 11) for Scalby instead of Bideford. Highlighting the 3 distinct zones of the parameter space isn't as clear at Scalby.*

[Figure]

*See also the below figure, an example of figure 11 (now figure 12) for Scalby. Because the y/FR trade off isn't as distinct as the one seen at Bideford. when observing the range in topographic profiles, while still showing the same relationship, isn't as clear. When y is increased, the platform still steepens, and when y is decreased, the platform becomes shallower, as it does at Bideford. Also, as this y/FR trade-off is found at the lower limits of y, the plot of the lower limits of y goes beyond our constrained parameter space.*

[Figure]

Figure 11: perhaps change the colour so that the two types of triangles are distinct?

*Triangles used the same colours before to clarify the points are equal +/- from the*
*regression line. Nevertheless, the colour of triangles has been changed as suggested*
*to make them more distinct.*

Part 5.2. Wave decay/Material resistance comparison. This is interesting, but the fact that a
lower wave erosive capacity has to be compensated by an increased erodibility to achieve
the same result is a bit trivial. On the other hand, I think we can go further. The dissipation
of wave energy should decrease exponentially across the width of the platform. I imagine
that the effect of a faster decay is not exactly compensated by an increased erodibility. For
example, high dissipation with low resistance should favour the erosion of the outer part of
the platform while low dissipation and high resistance should erode the inner part more
strongly. It might be possible to discriminate between the two components. Furthermore I
suspect that the best fit in figure 11a is not a straight line but a curved one.

*The reviewer makes an interesting and important point here, that we failed to focus*
*on in the previous discussion. We have therefore, restructured section 5.2 to explain*
*how both the spatial distribution and magnitude of the wave height decay rate can*
*relate to the material resistance. We also highlight how our results demonstrate*
*both of these behaviours, with the change in steepness in Fig. 12 and the*
*comparisons between Scalby and Bideford sites that exhibit dissimilar shore*
*platform gradients and as a result have different best-fit results for wave height*
*decay rate .*

*Line 1191-1260: "Nevertheless, the relationship between a and b is not as*
*straightforward as saying faster wave height decay needs to be compensated by a*
*lower material resistance. Varying the wave height decay rate (a) changes the*

*erosive energy distribution across the shore platform, and this ultimately influences the amount of erosion achieved by waves. When waves dissipate energy too quickly (a is increased), erosion of the outer part of the platform is increased and less erosion is achieved towards the cliff base. As a result, modelled topographic profiles become too steep to match the gradient of the shore platform measured at Bideford (Fig. 12c). In contrast, when waves dissipate too slowly (a is decreased) and waves dissipate energy across a wider distance of the shore platform, erosion is increased further inshore and overall erosion across the shore platform is increased. The gradient of the modelled topographic profiles become lower than measured at the Bideford shore platform (Fig. 12e). Here we demonstrate the twofold impact of varying wave height decay rate: 1) increasing the distance across which waves break, increases the amount of energy made available for erosion, and 2) varying the rate of wave dissipation affects the spatial distribution of erosion across the shore platform. The observed range of residuals across the b/a regression and the resultant model outputs highlights the narrow uncertainty of y required to produce a matching topographic and $^{10}Be$ concentration profile.*

*In order for an MCMC analysis to produce effective posterior distributions, the optimisation method requires free parameters to be independent of each other. As a result of the correlation revealed between a and b parameters, the high confidence placed on a values (Fig. 12) is not reflected by the posterior distributions produced from the MCMC results (Table 3). Wide posterior distributions of the accepted samples (axis histograms in Figure 12a) produce large uncertainty for final MCMC results. We argue that propagating MCMC uncertainties for a together with the uncertainty for b produces unrealistic errors in model outputs, specifically seen in the large range of shore platform gradients, because of the correlation between these two parameters. Consequently, the uncertainty on final model outputs (Fig. 6; Fig. 7) are constructed by plotting the model result of the median and 16-84% confidence range for each parameter against the median result for the other two parameters.*

*Comparisons between the two sites further support our observations of the relationship between material resistance (b) and wave height decay rate (a). The platform gradient at Scalby is shallower compared to Bideford, and best-fit results for a show wave dissipation needs to be slower to match the topographic profile (Fig. 10). Best-fit a values are constrained by the lowest bound of a for Scalby, where a limits are informed by field-based studies (Ogawa et al., 2011). For Scalby, this either means: 1) overall wave erosion needs to be greater, or 2) wave erosion needs to be more evenly distributed across the shore platform, compared to at Bideford. Furthermore, Scalby is located at a meso-tidal coastline with mean spring tidal ranges of 4.6 m, and previous studies have noted the positive correlation observed between platform gradient and tidal range for real-world sites (e.g. Matsumoto et al. (2017)). Future investigations into how b vs a relationships may change as a function of platform gradient and/or tidal range within this exploratory model that are informed with additional site-specific datasets are needed in order to understand this model behaviour further."*

Line 799 "provided wave height decay rate (a) has adjusted accordingly" I am not so certain
about that, refer to my previous remark.
*We agree again with the reviewers comments here, and believe we have*
*appropriately addressed their concern in additional text added (see above*
*comment).*
Lines 814-817. I fully agree with the authors.
*We are happy to see that the reviewer is in agreement with this discussion topic we*
*thought important to clarify.*

Reviewer #2 (Anonymous)

**1 Summary**

I appreciate the opportunity to review Shadrick and others (submitted to Earth Surface
Dynamics, 2021). In this study the authors estimate millennial scale cliff erosion rates using
a coupled model of rocky coast evolution and cosmogenic radionuclide production. The
authors implement a multi-objective optimization approach to understand the relative roles
of 10Be concentrations and topographic profiles on constraining model parameters.
Because the authors consider two sites, they are able to provide insight into the extent to
which results are general or site specific. The study is well designed and well described. The
work reflects a novel application of optimization to interpreting geologic data and it
represents a big advance in using this type of methodology to interpret geologic data and
geomorphic models. The authors nicely explore a variety of important topics including the
relative contribution of different types of data to parameterize a coupled model, and the
role of parameter covariance and equifinality.

I recommend it for publication after minor revision.
*We are delighted to see the reviewer provide an encouraging response to our work. We*
*are particularly enthused that the reviewer clearly sees the novelty and usefulness of*
*the model optimisation methodologies we have developed.*

**2    Narrative comments**

Provide more explanation and/or justification for the 8 kyr model duration. Was this chosen
because it is the extent of GIA modeling, needed for RSL as a boundary condition, or
because of some other reason? On this same point, I'd recommend providing a bit of
context to the reader regarding typical timescales to reach a steady state topographic
profile under steady forcing. It is clear that this is not your aim, since you use nonsteady RSL
forcing, and will of course depend on parameter values... however, it will be helpful context
for how far away from equilibrium the coastal profile would be over a 8 kyr duration.
*We agree with the reviewer here, that further explanation of the chosen model*
*simulation time is needed on reflection, and more so because similar comments*
*have been made by reviewer #1.*

*See example plot below that compares a 10,000-year simulation time to an 8,000-*
*year simulation time.*

[Figure]

*Line 362-367: "We chose to implement a model simulation time of 8000 years. A*
*simulation time of 8000 years BP to present day captures the RSL history curve for*
*both sites (Fig. 2), where rapid RSL rise occurs for the first ~1000 years, followed*
*by a slow decline from 7000 years BP to present day. So, we can observe how cliff*
*retreat rates will respond to these different stages in the RSL history. Having tested*
*longer simulation times, implementing a simulation time of 10000 years, for*
*example, would show no change to final model outputs for nearshore topography*
*or 10Be concentrations. Our results are thus independent of this initial boundary*
*condition, and longer simulations would increase the computer run time*
*unnecessarily.*

Similarly, provide additional information regarding what initial conditions were used in the
model, and whether the model shows sensitivity to the initial conditions over the timescale
simulated.

*Initial conditions are included in a table in the supplementary materials, we have*
*added a reference to this and have made sure to highlight that initial conditions*
*have little impact on the long-term trajectory of retreat rates and endmember*
*topographic and CRN profiles.*

*Line 352-355: "Other fixed model parameters and initial model conditions are set*
*to the same values as used by Matsumoto et al. (2018) (supplementary materials*
*Table S7). Once the model burn-in period has been completed, (first ~1000 years),*
*the initial conditions, such as platform gradient, have negligible effect on final*
*outputs of topography, $^{10}$Be concentrations and retreat rates."*

Recommend that most of section 3.4 (MCMC inputs) be moved into section 3.2 (The coastal
evolution model). Specifically, I would recommend adding to section 3.2 an introduction to
the parameters y and K (as FR is already introduced) as well as a brief description of other
parameters in the model presented by Matsumoto, Dickson, and Kench (2016) which are
not considered here. It is certainly reasonable to not consider these parameters, but let the
reader know a bit more about what they are. I would then recommend putting most of section 3.4 at the end of a revised section 3.2. This will help the reader understand what the
model and parameters are before you begin section 3.3 and discussion of Dakota.
*We thank the reviewer for this helpful suggestion, to make the complex explanation*
*of model background and parameters clearer. Following these, we have*
*restructured these sections as suggested and added a schematic diagram to help put*
*model parameters in context of other model parameters involved. We think we have*
*given enough context of other model parameters for the optimisation uses here.*
*See below the added schematic figure:*

[Figure]

*Line 339-344: "Figure 3: Coastal evolution model schematic. Topographic profile*
*cross-section constructed in a gridded framework, showing wave approach and*
*influence of tidal duration distribution. MHWS, MHWN, MT, MLWN, MLWS*
*denote mean high water spring, mean high water neap, mid tide, mean low water*
*neap and mean low water spring. Binary values of 0 and 1 are assigned to water/air*
*and rock categorised cells respectively. Rock cells (value 1) are eroded and removed*
*from the profile (assigned value 0) once wave force exceeds material resistance ($F_W$*
*$\geq F_R$) (b,c). Subaerial weathering (K), can also lower the material resistance value*
*($F_R$). Wave height decay rate (y) controls the distance waves can break across the*
*shore platform and as a result, the erosional potential of wave assailing force $F_W$."*
I was surprised not to see a plot of the pareto front itself (that is, a plot with the scaled and
weighted topographic RMSE on one axis and the 10Be RMSE on the other axis, with one line
for each of the two sites). The simulated topography and 10Be concentrations are provided
in Figure 4, but the pareto trade-off itself is important to visualize in a multi-objective
function study. As the multi-objective nature of this study is so novel, I'd strongly recommend adding such a subplot to Figure 4. It would support the text from lines 435–448,
namely that the hump in the Bideford 10Be data is best fit only when the topographic fit is
mostly ignored.

*We thank the reviewer here for highlighting our missed opportunity to visualise the*
*pareto front and agree strongly that a subplot to figure 4 will greatly improve this*
*figure and findings. Subplot has been added to figure 4 and figure caption updated.*
*Additional text has also been added to explain the pareto front addition to the figure*
*and provide interpretation.*

[Figure]

*Line 776-789: "Figure 5: The five Pareto set results for both Bideford (a) and*
*Scalby (b) sites. The modelled topographic profile and $^{10}$Be concentration profile*
*are shown alongside corresponding measured data. Modelled cliff retreat rates are*
*shown for the past 7000 years. Yellow coloured model results correspond to 50 –*
*50% objective function weighted MCMC results. Darkest blue coloured model*
*results correspond to the MCMC results that were most weighted towards the*
*topographic (Topo) profile (95%). Darkest red coloured model results correspond to*
*the MCMC results that were most weighted towards the $^{10}$Be concentration (CRN)*
*profile (95%). The Pareto front of scaled and weighted $^{10}$Be and topographic*
*objective functions is shown for both sites (c)."*
*Line 752-765: "For the Pareto front, where the scaled and weighted topographic*
*and $^{10}$Be concentration objective functions are compared, the sensitivity of different*
*weighting sets to final model results at Bideford is revealed (Fig. 5c). The Pareto set*

*at Bideford again suggests we should weight more towards the topography, but only*
*when we weight the combined objective function 95% towards the $^{10}Be$*
*concentration profile RMSE, do we see a poor match to the topography (Fig. 5a).*
*In contrast, at Scalby, all combinations of objective function weightings produce*
*very similar model outputs (Fig. 5b). This reveals that the best-fit model result for*
*the topographic profile and the $^{10}Be$ concentration profile are found in the same*
*parameter space for Scalby, but not necessarily for Bideford. Uniformity in results*
*across the Pareto set for Scalby is further supported by the Pareto front (Fig. 5c).*
*For Scalby, the Pareto front shows the expected, convex shape of a Pareto front*
*that looks to minimise both objective functions simultaneously.*
*Crucially, final results from the 50 – 50% weighted MCMC analysis show a good*
*representation of the full Pareto set of output model result (Fig. 5)."*
None of the simulations seem to capture well the increase in slope in the most shore-ward
50 m of the the Scalby site. Could you discuss this more? Are there field observations from
Scalby that might provide more context to this topographic feature?
*We agree with the reviewers observation here, that the nearshore topographic slope*
*is not matched at Scalby. The transect of CRN samples taken from the shore*
*platform at Scalby had to move between sandstone beds to avoid local erosion spots.*
*The dsm-extracted topographic profile therefore is calculated from sample locations*
*and distances from the cliff-platform junction are superimposed onto this transect.*
*The location of this transect includes a part of the beach profile therefore and here*
*is where we see the increase in topographic profile nearshore. Statement added:*
*Line 795-796: "The nearshore increase in the measured platform slope seen at*
*Scalby is a result of boulder accumulation near the cliff foot."*
Your results show that cliff retreat rates match RSL rise closely. But they also show (Figure 6)
that scaled RSL is on the lower end of the Bideford retreat rates, while it is on the upper end
of the Scalby rates (dashed line is below(above) the mean line for Bideford (Scalby)). Is this
an interpretable result? 1 Are there differences in the wave climate, geomorphology, or
geology of the sites that could explain this?
*Although an interesting observation, we don't think this is an interpretable result as*
*although cliff retreat rates at both sites are closely tied to RSL, differences relative*
*to one another relate to site specific factors such as lithology and/or wave energy*
*balance. Both of which we are unable to comment on in this study without further*
*site-specific data/ observations. Nevertheless, we will look into this further in the*
*secondary publication where we may be able to interpret these results in relation to*
*site specific observations.*
You nicely discuss parameter covariance near the end of the manuscript. However, there
were two points related to parameter covariance that I think are important to discuss. First
is related to your specific implementation. You set K = 5cFR = 5c10b , removing some
parameter covariance that would have existed otherwise. Why not treat them as fully
independent and document the nature of the covariance as you do for a and b in section
5.2.

*We thank the reviewer for an interesting comment, and one we have actually*
*explored previously in test simulations. We already know that a clear linear*
*relationship is exists between material resistance and weathering rate: as material*
*resistance increases, weathering rates need to proportionately increase to produce*
*the same model output. Following Matsumoto et al. (2018), we varying K as a*
*function of FR, as this is the primary control on the style of model evolution. As FR*
*is an abstractly defined representation of rock strength, varying weathering rates as*
*a proportion of the material resistance seemed to provide more meaningful results.*
*Furthermore, in preliminary test simulations, we initially did allow K to be*
*independent of FR, but observing the objective function surface showed insightful*
*results, to what we already expected and introducing the third free variable (y), with*
*K independent of FR complicated the objective function surface unnecessarily.*

*Line 596-597: "Following Matsumoto et al. (2018), maximum intertidal weathering*
*rate (K) is varied as a proportion of the material resistance, in order to capture*
*controls on the variation in topographic development."*

A second, bigger picture comment that you are well poised to make is about the "meaning"
of these model parameters. Namely, when model authors write models, we often think of
the parameters as independent and meaningful—and sometimes linked to field or
laboratory measurements (Dietrich et al. 2003). But it is not uncommon to find these sort of
covariance issues, most often because of how these parameters are used. In the case of the
model presented by Matsumoto, Dickson, and Kench (2016) is this something that could be
anticipated because of the mathematical form of the model? Or is it fully emergent.

*The reviewer highlights an interesting discussion point. However, due to the*
*abstract representation of processes used in this coastal evolution model, the*
*covariance seen between parameters is not unexpected. This understanding of*
*parameter covariance has been developed through previous exploration studies*
*using this model e.g. Matsumoto et al. (2016, 2018).*

One theme that emerges from both sites is the overwhelming influence of the RSL boundary
condition. Looking at the different cliff retreat rates in Figure 4 it seems reasonable to
conclude that no matter the topography vs 10Be weighting the estimated cliff retreat rates
match the RSL forcing. It might be worth commenting in the discussion on the relative role
of the boundary condition vs parameter values for this type of analysis.

*We agree strongly with the reviewers comment here and have added a relevant*
*comment to the end of the discussion section 5.4. Again, we want to leave detailed*
*discussion on this topic for the secondary paper.*

*Line 1320-1322: "Furthermore, consistent trends in past cliff retreat rates for all*
*Pareto weighting sets (Fig. 5), that match the declining rate of RSL rise, suggest the*
*influence of the RSL boundary condition dominates over individual parameter*
*values in this model."*

Overall all figures are well designed and clear. However, my black and white printer led to
this comment: consider a grayscale safe color scales (e.g., viridis used in Fig 9) and/or
different symbols for Bideford and Scalby sites. Colorbrewer is a good resource for this.

*Shaded areas darkened and grey colour changed in figure 5.*

**3 Line level comments**

Bullet points in this Section indicate "<LineNumber>","T<Table Number>", or "F<Figure Number>".

Perhaps add a statement about the sort of timeframes that would be possible without this method. Such a statement would make clear by contrast the benefit of this approach.

*We thank the reviewer for highlighting this point that emphasises the importance and significance of this work in relation to the long-term timescales we're able to model. We do comment on the short timescales only available without use of CRN application and long-term modelling in the introduction, but do agree that the abstract is strengthened with a statement of it here. Adding this additional information to the abstract strengthens the novelty and purpose of developing these rigorous optimisation methods.*

*Line 15-16: "Without such methods, long-term cliff retreat cannot be understood well, as historical records only cover the past ~150 years."*

The model represents a cross-section, yes? Recommend using the term "cross section" if this is the case.

*Yes, we are modelling a cross-section of the cliff and shore platform topography, or a transect taken perpendicular from the cliff line. We thank the reviewer for signalling an improvement could be made in the clarity of our explanation. Following the reviewer's suggestion, we have described the topographic profile as a cross-section and have used this term throughout the manuscript:*

*Line 189-190: "The exploratory model uses a grid framework, in which cells are assigned a binary value of 1 (rock) or 0 (water/air), and represents a cross section transect (elevation and distance), taken perpendicular to the cliff line (Fig. 3)."*

*Line 390-392: "In this study, we use the coupled model to simulate both a topographic profile and also a $^{10}$Be concentration profile. The first model output is the cliff-platform profile, which displays a cross section of the elevation, width and gradient of the modelled shore platform in an across-shore orientation."*

At the end of this paragraph I wanted a sentence or two introducing the concept of the pareto front.

*Although we have a separate section that explains the basic concept of the pareto front, we have added a few sentences here, as suggested by the reviewer and point to the section describing the pareto front in more detail.*

*Line 381-383: "Multiple MCMC simulations are performed, each with different weightings assigned to the topographic profile and $^{10}$Be concentration profile to construct a Pareto front of optimised results across the range of weightings explored (see section 3.3.3)."*

I think you need to add something like the phrase "with different weights" to the end of this sentence, because it is specifically the nature of the different weights that allows you to explore the pareto front.

*We thank the reviewer for this suggestion and have added a reference to the*
*different weights to this sentence.*
*Line 398-399: "Multi-objective optimisation is used to find a set of model input*
*parameters that minimises both topographic and $^{10}$Be concentration residuals with*
*different weights."*
235 I find it helpful to include a section or subsection "Model Implementation" that puts
together information like the cell size, as well as a few items not stated. For example, what
timestep was used? This would also be where you might state the simulation duration and
why it was chosen.
*In structuring the methodology section into chosen subsections, we attempted to*
*simplify all the components of the optimisation method as much as possible. From*
*both reviewers comments it is clear that we can do more to help readers follow the*
*methods descriptions as clearly as possible. We have therefore taken the reviewers*
*advice, to add another subsection entitled 'model implementation' to include*
*information asked for here, and by reviewer #1, regarding timestep etc.*
*Line 351-374: "3.2.1 Model implementation*
*Other fixed model parameters and initial model conditions are set to the same*
*values as used by Matsumoto et al. (2018) (supplementary materials Table S7).*
*Once the model burn-in period has been completed (first ~1000 years), the initial*
*conditions, such as platform gradient, have negligible effect on final outputs of*
*topography, $^{10}$Be concentrations and retreat rates. The RSL history input is taken*
*from the GIA model of Bradley et al. (2011). RSL uncertainty was not considered as*
*we expect it to make little difference to final results. For southern UK sites across*
*the late-Holocene, the misfits between measured RSL data and GIA model*
*predictions are minor (Bradley et al., 2011). Uncertainties of magnitude $\pm0.01$-$0.1$*
*mm y$^{-1}$ of RSL rise have negligible impact due to the spatial and temporal*
*resolution considered for the model. A fixed mean spring tidal range of 8.41 m for*
*Bideford and 4.6 m for Scalby are used, which are based on tide gauge records*
*(National Tidal and Sea Level Facility, 2021).*
*We chose to implement a model simulation time of 8000 years. A simulation time of*
*8000 years BP to present day captures the RSL history curve for both sites (Fig. 2),*
*where rapid RSL rise occurs for the first ~1000 years, followed by a slow decline*
*from 7000 years BP to present day. So, we can observe how cliff retreat rates will*
*respond to these different stages in the RSL history. Having tested longer*
*simulation times, implementing a simulation time of 10000 years, for example,*
*would show no change to final model outputs for nearshore topography or $^{10}$Be*
*concentrations. Our results are thus independent of this initial boundary condition,*
*and longer simulations would increase the computer run time unnecessarily.*
*Modelling rock coast evolution across an 8000-year window means only a*
*Holocene history for shore platform formation has been considered, with no*
*possible re-occupation from a previous interglacial period (e.g. Choi et al. (2012)).*
*The $^{10}$Be concentration datasets used to develop this optimisation routine at both*
*sites exhibit low concentrations, suggesting these rock coast features are Holocene-*
*formed (Regard et al., 2012). Therefore, these datasets are suitable for modelling*
*Holocene-formed shore platforms, as a means to develop this optimisation routine.*
*During the 8000-year simulation time, the topographic profile and $^{10}$Be*

*concentrations are calculated and output every year (1-year timestep). The model*
*space is split into 10x10cm gridded cells (Fig. 3)."*

242 You mention measurement error here, but it is not clear whether measurement error
was incorporated into the RMSE. In your case, I'd expect you could have a different
measurement error for each 10Be observation, and that including/excluding this might have
an impact on the Bideford results because the highest 10Be measurements also have the
highest error.

*In this case, individual measurement errors associated with each measured data*
*point were not included in RMSE calculations and so, all given datapoints are*
*given equal weightings. We agree with the reviewer here, in that because highest*
*$^{10}$Be concentrations have the highest measurement error at Bideford, this could*
*have an impact on the best-fit results, if measurement error had influence on the*
*weightings of each datapoint in the RMSE calculation. For example, in the case at*
*Bideford, essentially weighting each $^{10}$Be concentration by the measured error*
*would result in the least weighting being given to the highest concentrations at the*
*peak of the distribution. However, because the best fit results at Bideford match the*
*lower $^{10}$Be concentration samples better that the highest, at the peak of the*
*distribution, for this example we believe including measurement errors will have*
*little impact on the final result. Each datapoint for the topographic profile also has*
*the same measurement error. We have added another sentence to clarify that we*
*have not included measurement error.*

*Line 417: "In this case, we have not considered individual datapoint measurement*
*errors in the RMSE calculation."*

246 Why is $w_i$ in a square root? I would expect just $w_i$ here such that $\sum_{i=1}^{N_i} w_i = 100$.
*Dakota multiplies the scaled residual by the square root of $w_i$ as this function is*
*squared within the likelihood function. For clarity, we have edited the formula here*
*to show the combined objective function as the likelihood calculation used in*
*Dakota. We have formulated this in a way that the squared root weighting has*
*already been squared so readers understand the weighting to be applied in the way*
*the reviewer has described above.*

*Line 425: "$Likelihood_p = \prod_{i=0}^{N_i} \frac{1}{\sqrt{2\pi}} exp\left[-\frac{w_{i,p}\left(\frac{RMSE_i}{s_i}\right)^2}{2}\right]$* (2) "

255 Your discussion and approach to scaling makes sense when I read it here. However,
after reading this portion of the methods I was confused when I got to Figures 7, 8, and 9
because I was expecting the topographic and 10Be values to be of similar magnitude (which
they are not). I think my confusion could be addressed with minimal revision to sections
3.3.2 and 3.3.3. Specifically, I would recommend talking only about constructing the topo
and 10Be RMSE values in the first of these sections and introduce an equation that looks
something like

$$RMSE_i = \sqrt{\frac{1}{N_j} \sum_{j=1}^{N_j} \left(\frac{Mod_{i,j} - Meas_{i,j}}{\sigma_{i,j}}\right)^2} \qquad (1)$$

Here I've also added a term for measurement error $\sigma_{i,j}$. Then, after discussing how the two

RMSEs were each constructed, in the second section introduce scaling/weighting and constructing the pareto front with an equation like

$$TotalRMSE_p = \sum_{i=0}^{N_i} \frac{w_{i,p}}{s_i} RMSE_i \qquad (2)$$

(2) This would more clearly separate the construction of your two objective functions from the scaling and weighting of them for the pareto analysis since you don't need to discuss scaling/weighting until you get to the discussion of multiobjective and the pareto front.

Similarly, I would recommend adding a subscript of some sort (I've used p above) to denote that the weights change depending on which pareto set is being used.

*We have separated equation into 2 separate equations as suggested, but as we have*

*not included measurement error of individual data points, we have not included the*

*measurement error addition to equation 1, as suggested.*

*Line 401-436: "First, the root mean square error (RMSE) is calculated both*

*between the modelled and measured DSM-extracted topographic profile and also*

*the modelled and measured $^{10}$Be concentration profile, respectively. Modelled*

*outputs and measured data are shifted to the final (present-day) modelled cliff*

*position, where the final cliff position is at 0m. Interpolation is used to assign*

*corresponding modelled data (cell resolution = 0.1 m) to every measured data*

*position across the shore profile. For every measured data point, the elevation and*

*concentration residuals are calculated and combined into a RMSE score for both*

*topographic and $^{10}$Be concentration model outputs:*

$$RMSE_i = \sqrt{\sum_{j=1}^{N_j} \left(\frac{Mod_{i,j} - Meas_{i,j}}{N_j}\right)^2} \qquad (1)$$

*In Eq. (1), for each objective function i, the residuals (Mod_{i,j} – Meas_{i,j}) are*

*calculated between the modelled and measured data values, which are indexed by*

*subscript j. The number of measured data points are distinct to the topographic*

*profile and $^{10}$Be concentration profile datasets and are denoted by $N_i$.*

*Next, both RMSE values are then scaled ($s_i$) within Dakota to 1) equalise the*

*magnitude ranges of both the topographic and cosmogenic radionuclide RMSE*

*scores, and 2) set the RMSE magnitudes to a sensible multiple relative to the*

*default measurement error used by Dakota in the likelihood function: variance is*

*assumed to be 1.0 when no measurement error is specified. In this case, we have*

*not considered individual datapoint measurement errors in the RMSE calculation.*

*As a result, scaled RMSE scores for both the topographic and $^{10}$Be concentration*

*profiles are within the range of ~0 to 10. Individual weightings ($w_i$) are applied to*

*the scaled RMSE functions for both the topographic and $^{10}$Be concentration*

*profiles (Adams et al., 2019). The scaled and weighted RMSE scores are combined*

*within a Gaussian likelihood function, and the final composite objective function, Likelihood_p, becomes:*

$$Likelihood_p = \prod_{i=0}^{N_i} \frac{1}{\sqrt{2\pi}} exp\left[-\frac{w_{i,p}\left(\frac{RMSE_i}{s_i}\right)^2}{2}\right] \qquad (2)$$

*In Eq. (2), $N_i$ is the number of individual objective functions we aim to collectively minimise. In this case, we have two individual objective functions ($N_i = 2$): a topographic profile and a $^{10}Be$ concentration profile. Future applications may add additional objective functions ($N_i > 2$), for example, a secondary CRN concentration profile (e.g., $^{26}Al$ or $^{14}C$). Weightings applied to the separate RMSE scores are denoted by $w_i$, where subscript i refers to specific values associated with each individual objective function. The weightings applied to the topographic profile and $^{10}Be$ concentration profile are changed between MCMC inversion calculations in order to construct the Pareto set of optimised results (see section 3.3.3). The scaling values are denoted by $s_i$ and are exclusive to the individual objective function. A topographic profile scaling value is calculated by summing the standard error from a linear regression of the topographic profile and the resolution of the UAV imagery. The average measurement error of $^{10}Be$ concentrations for each site is used as a scaling value for the $^{10}Be$ profile. Table S7 in supplementary materials summarises the objective function scaling values for both sites. Subscript p refers to the different set of weights ($w_{i,p}$) assigned to each objective function ($RMSE_i$) used to construct the pareto front. "*

Recommend introducing y and K earlier, as discussed in the narrative comments.

*As indicated in above responses, we have moved details from section 3.3.4 MCMC analysis of model parameter inclusions in section 3.2 The coastal evolution model.*

*Line 195-200: "Erosion achieved by breaking and broken waves can be changed by varying the distance across the shore platform that waves can dissipate energy: wave height decay rate (y) (Fig. 3). A small value for y means wave height will decay slowly, in which case breaking waves exert energy across a greater distance of the shore platform surface, which achieves more erosion. In contrast, a large value for y indicates that wave height will decay quickly and wave-driven erosion covers a shorter distance across the shore platform."*

*Line 204-206: "The conceptual value for material resistance ($F_R$) is highly simplified by incorporating mechanical, geological and structural rock factors into a single value to represent rock mass strength (Matsumoto et al., 2016)."*

*Line 210-211: "Maximum weathering rate (K) occurs at the mean high water neap tidal level (MHWN), which is defined by a weathering efficacy distribution (Porter et al., 2010) (Fig. 3)."*

This is the first point in the text where the simulation duration of 8 kyr is mentioned. I would recommend mentioning it earlier as well as providing context regarding why that duration was used at that point in the text.

*Again, we have now included this earlier in the new addition of the model implementation section.*

458 Since you only consider one point on the pareto front, I think this section title is
misleading. Consider revising.
*As suggested, this subtitle has been changed from 'Model results from multi-*
*objective optimisation' to 'Best-fit model results'.*
*Line 789: "4.2 Best-fit model results"*
519 The last sentence of the figure caption is confusing to me. I think you mean to say that
the gray shaded region corresponds to cliff retreat rates which occurred when the cliff was
further offshore than your topographic profile measurements. Revise for clarity.
*Wording changed for clarity*
*Line 870-871: "The grey shaded area corresponds to the cliff retreat rates which*
*occurred further offshore than where the measured data was collected."*
683–685 Recommend framing this differently. Rather than making a statement about
confidence, I'd recommend pointing out that it is rare to formally evaluate how different
data sources (10Be, topo) constrain a model differently. Each of these are valid data, and if
you had only one (most commonly, no 10Be) it would be totally fine to parameterize the
model with only the available data (see, for example, any paper fitting river long profiles to
some sort of fault history and/or value of fluvial erosion coefficient). But because you have
both types of data you have the opportunity to evaluate the relative information provided
by each data source. If, as is not the case here, both data sources yielded the same
parameter estimates, we would learn we don't gain anything from the second dataset (and
it is thus not necessary to make those observations). In contrast, as you find, if the two data
sources contain different information, these datasets pull the coupled model in different
directions. An potentially relevant reference here is Furbish (2003).
*We thank the reviewer here for their suggestions, and we agree that reframing this*
*discussion is appropriate. See revisions below:*
*Line 1078-1082: "In this study, we have a rare opportunity to formally evaluate*
*how two distinctive datasets constrain a model differently. We find the two datasets*
*used here reveal dissimilar patterns in the objective function space between the*
*topographic profile RMSE and the $^{10}$Be concentration RMSE (Fig. 8-10). The*
*topographic data and $^{10}$Be concentration data have therefore, provided us with*
*different information and validates the use of multi-objective optimisation in*
*understanding the long-term evolution of rock coasts."*
T1 Are the weights listed $w_i$ or $\sqrt{w_i}$ . Please clarify.
*See response to comment on line 849 of this response document.*
T2 Why is the base for K 5 rather than 10?
*Following Matsumoto (2018) implementation of weathering rate (K). We wanted to*
*be consistent with how the 3 free parameters are varied within the parameter space.*
F3 I'd recommend adding one more loop here to denote that this workflow is done for each
location on the pareto front. You nicely emphasize how your approach could be generalized to include additional objective functions, and this could help emphasize that it can be
generalized to additional points on the pareto front.
*Additional loop added to figure and figure caption updated.*

[Figure]

*Line 514-518: "Figure 4: Structure for implementing a single MCMC calculation*
*using Dakota. Data inputs into the coupled model include a topographic profile, a*
*[10]Be concentration profile and a RSL history. The MCMC analysis is performed*
*multiple times with different weightings (shown by the blue loop) for the objective*
*functions (topographic profile RMSE and [10]Be concentration profile RMSE) and*
*produces a corresponding maximum likelihood estimation (MLE\*) result. For each*
*MCMC calculation, the Weights\* value is changed for each RMSE score. The*
*different values for the Weights\* are shown in Table 1 and correspond to $w_i$ (Eq. 1).*
*The set of MLE results together produce the 'Pareto front' of multi-objective*
*optimised results."*
F5 Because the right column represents a zoom into a portion of the left column, a gray
rectangle or similar that indicates this region in the left column would guide the reader.
*Grey box added to figure 5 to indicated zoomed area.*
*Line 838: "Grey boxes (a,c,e,g) correspond to 300 m distance offshore (b,d,f,h)."*
F6 Similarly a box in the upper right of each panel showing the extent of the inset would be
helpful.

*As the non-grey shaded area in the main plot corresponds to this inset, adding*
*another box to indicate the extent of the inset again would be unnecessary.*
*However, we have changed the figure caption to make this clearer:*
*Line 870-871: "The grey shaded area corresponds to the cliff retreat rates which*
*occurred further offshore than where the measured data was collected, and the*
*non-shaded area shows the extent of the inset plots."*
F6 Why only show 7 kyr when the simulation duration is 8 kyr. If the first 1 kyr is a "burn in
period" to forget the initial conditions, that is reasonable but should be stated.
*Added statement to figure caption to explain first 1000 years is a burn in period.*
*Line 865-866: "The first 1 kyr is excluded as this corresponds to the burn in period*
*of the model."*
F7,8,9 Rather than plotting the objective function itself in Figures 8 and 9 I would
recommend plotting the posterior. It is often easier to interpret because it does not have
the same issues with overplotting that the objective function provides. One nice tool for this
is corner.py (Foreman-Mackey 2016). It nicely also shows the marginal distributions.
*As discussed briefly in discussion section 5.2, we chose not to show/plot the*
*posterior distributions of the free parameters as the correlation between parameters*
*cause there to be wide histogram distributions which were not very useful. We tried*
*multiple methods of displaying the final results, including python seaborn pairplot*
*(similar to corner.py), but believe the simple scatter of the different objective*
*function shows the results in the simplest and easily digestible way. Also, by*
*observing the scatter for all the tested samples for the topographic RMSE and $^{10}$Be*
*RMSE, and then only the accepted samples for the combined likelihood, nicely*
*shows how both objective functions are combined and how the MCMC algorithm*
*accepts/ rejects samples based on their combined misfit to the measured data.*
F10 It is not clear if these three different sets (orange, blue, pink) come from different
pareto sets, or different samples from the 50-50 evaluation. Clarify.
*Statement added to figure caption:*
*Line 1178: "Model results are from the 50 - 50% weighted Pareto set simulation."*
756 This is a very nice subsection and analysis.
*We are delighted that the reviewer finds interested in this discussion analysis. We*
*feel that the addition of suggestions by reviewer #1 strengthens this subsection*
*further.*